# Aberrant methylation underlies insulin gene expression in human insulinoma

Esra Karakose[1,6], Huan Wang [2,6], William Inabnet[1], Rajesh V. Thakker [3], Steven Libutti[4], Gustavo Fernandez-Ranvier [1], Hyunsuk Suh[1], Mark Stevenson [3], Yayoi Kinoshita[1], Michael Donovan[1], Yevgeniy Antipin[1,2], Yan Li[5], Xiaoxiao Liu [5], Fulai Jin [5], Peng Wang [1], Andrew Uzilov [1,2], Carmen Argmann [1], Eric E. Schadt [1,2], Andrew F. Stewart [1,7 ✉], Donald K. Scott [1,7] & Luca Lambertini [1,6]

Human insulinomas are rare, benign, slowly proliferating, insulin-producing beta cell tumors that provide a molecular "recipe" or "roadmap" for pathways that control human beta cell regeneration. An earlier study revealed abnormal methylation in the imprinted p15.5-p15.4 region of chromosome 11, known to be abnormally methylated in another disorder of expanded beta cell mass and function: the focal variant of congenital hyperinsulinism. Here, we compare deep DNA methylome sequencing on 19 human insulinomas, and five sets of normal beta cells. We find a remarkably consistent, abnormal methylation pattern in insulinomas. The findings suggest that abnormal insulin (INS) promoter methylation and altered transcription factor expression create alternative drivers of *INS* expression, replacing canonical *PDX1*-driven beta cell specification with a pathological, looping, distal enhancer-based form of transcriptional regulation. Finally, NFaT transcription factors, rather than the canonical *PDX1* enhancer complex, are predicted to drive *INS* transactivation.

[1] From the Diabetes Obesity and Metabolism Institute, The Department of Surgery, The Department of Pathology, The Department of Genetics and Genomics Sciences and The Institute for Genomics and Multiscale Biology, The Icahn School of Medicine at Mount Sinai, New York, NY 10029, USA. [2] Sema4, Stamford, CT 06902, USA. [3] The Academic Endocrine Unit, University of Oxford, OX3 7LJ Oxford, UK. [4] The Cancer Institute of New Jersey, New Brunswick, NJ 08901, USA. [5] The Department of Genetics and Genome Sciences, Case Western Reserve University, Cleveland, OH 44106, USA. [6]These authors contributed equally: Esra Karakose, Huan Wang, Luca Lambertini. [7]These authors jointly supervised this work: Andrew F. Stewart, Donald K. Scott. ✉email: andrew.stewart@mssm.edu

Type 1 and type 2 diabetes affect 420 million people globally[1]. Both result from reduced numbers of normally functioning insulin-producing pancreatic beta cells, a consideration that has led to attempts to develop drugs capable of inducing human beta cell regeneration. This has proven a challenge because adult human beta cells are terminally differentiated and recalcitrant to cell cycle re-entry. Encouragingly, we and others have identified drugs that are able to induce adult human beta cells to regenerate at potentially therapeutic rates[2–6], yet additional and more beta cell-specific regenerative drugs are needed.

Two rare diseases serve as models for human beta cell regeneration. One is the Beckwith–Weidemann syndrome (BWS)[7], which is associated with the Focal Variant of Congenital Hyperinsulinism (FoCHI)[8]. In this rare genetic syndrome, focal (mosaic) overgrowth of pancreatic beta cells results from imprinting abnormalities, maternal chromosomal loss, paternal uniparental disomy, and/or mutations in 11p15 region of chromosome 11. Beta-cell overgrowth and inappropriate insulin over secretion result in life-threatening hypoglycemia, which may require surgical resection of the hyperplastic region of the pancreas. A second example is provided by rare patients with insulinomas[9]. These are typically small (1–2 cm³), slowly proliferating, benign, beta-cell tumors that overproduce insulin, thereby causing hypoglycemia, seizures, and/or episodes of unconsciousness. Once diagnosed, insulinomas are usually easily removed by minimally invasive surgery. In terms of nomenclature, insulinomas comprise a subset of a larger class of pancreatic neuroendocrine tumors (PNETs). Some PNETs are "functional": they overproduce hormones such as insulin, gastrin, somatostatin, glucagon, and/or others[9]. Most PNETs, however, are non-functional (i.e., they do not overproduce or secrete hormones). By definition, if a PNET produces insulin in sufficient amounts to cause hypoglycemia, that PNET is considered an "insulinoma"[9]. The only normal human cell type that produces insulin is the pancreatic beta cell. Analogously, the only abnormal cell type that produces insulin is the insulinoma cell. Thus, both beta cells and insulinomas overexpress the *INS* gene, overproduce and oversecrete insulin, and they do this to approximately comparable degrees[9].

We have explored the genomics and transcriptomics of human insulinomas, hoping to reveal molecular or genetic pathways that can serve as additional drug targets for the induction of human beta cell regeneration for diabetes. In our initial studies of 38 sporadic human insulinomas[10], we observed three recurring signatures. The first was an epigenetic signature, evidenced by single-nucleotide variants (SNVs), multiple nucleotide variants (MNVs), insertions, or deletions (Indels), and/or copy number variants (CNVs) affecting genes involved in epigenetic control. Thus, although most insulinomas had mutations in different genes, 90% displayed variants in members of the Trithorax Group (TrxG) (exemplified by *MEN1*, *KDM6A*, *MLL3/KMT2C*), the Polycomb Repressive Complex (PRC) (exemplified by *EZH2*, *YY1*, *H3F3A*) or related epigenetic modifiers (exemplified by *KDM5B*, *KMT2C*, *SMARCC1*). These events were strongly associated with mis-expression of a broad range of TrxG- and PRC-regulated transcripts.

The second was a differential gene expression signature that included cell cycle regulatory genes and other genes whose misexpression appeared to result from variants in Trithorax, Polycomb, and related chromatin-modifying genes[10].

The third signature involved broad hypomethylation of CpGs in the well-known imprinted region spanning a portion of 11p15.5-p15.4[10]. In our earlier study, preliminary bisulfite DNA sequencing was performed on only two sets of purified beta cells and only ten insulinomas.

Since these abnormalities were quantitatively striking and clearly relevant to beta-cell proliferation and insulinoma pathogenesis, we have expanded these studies. Here, we describe deep methylome sequencing and analysis of each of the ~30,000 CpG dinucleotides within the 11p15.5-p15.4 region (referred to hereafter as "the 11p15.5-p15.4 target sub-region") in a statistically meaningful cohort of five sets of normal human beta cells and an expanded set of 19 insulinomas. We integrate and correlate the methylome results with additional large beta cell-relevant betacell data sets. The overall strategy employed in this study, as well as an illustration of sample source and use, is provided in Supplementary Figs. 1 and 2.

The result is a detailed study of the 11p15.5-p15.4 imprinted target sub-region in normal and abnormal cell types. It also provides a large and deep next-gen bisulfite sequencing data set in human beta cells. We find evidence of widespread, yet consistent and recurrent, methylation abnormalities throughout the target region, predicting altered promoter/enhancer usage in all human insulinomas, independent of their mutational signatures. The findings suggest that disordered 3D structural abnormalities in chromatin looping are common in, and contribute pathogenically to, the development of human insulinomas and their inappropriate insulin over secretion.

## Results

**Deep bisulfite sequencing in beta cells vs. insulinomas**. Adult human islets from seven normal human donors were obtained and beta cells isolated as described in "Methods" and Supplementary Fig. 1 and Supplementary Data 1. Three beta cell DNA samples were pooled to achieve the amount of DNA needed for the bisulfite sequencing, and four others were sequenced individually. The mean age of the four individual donors was 41.8 ± 5.4 (mean ± SEM), range: 20–69; BMI 26.7 ± 1.6; three were male, one was female. Human insulinomas were obtained at the time of surgery from 19 people, as described in Supplementary Data 2. The mean age was 47.9 ± 1.0 (mean ± SEM), range: 11–83; 12 were male and 7 were female. The size of the insulinomas ranged from 1.0 to 2.8 cm in greatest diameter, and Ki-67 labeling ranged from <1 to 20%. All but one were classified as "benign". Each displayed symptomatic hypoglycemia that was corrected by surgical removal of the insulinoma. Insulinomas from subjects known to be members of *MEN1* kindreds were intentionally excluded, although one insulinoma (Ins_27) was derived from a subject with a *MEN1* mutation.

DNA was extracted, and targeted deep sequencing of each of the 30,665 CpG dinucleotides in the 11p15.5-p15.4 target subregion was performed on the beta cells and the insulinomas as described in Methods. Briefly, the target sub-region of 1.35 Mbp spans coordinates 1,850,000–3,200,000 (NCBI37/hg19). This region extends telomerically from gene *SYT8* (which, in human islets treated with high glucose, generates DNA loops with the *INS* promoter[11–13]), centromerically to *OSBPL5* (the last imprinted gene in the 11p15.5-p15.4 region). Our choice of this target region was driven by its well-known imprinting abnormalities in FoCHI[8], BWS[7], and insulinoma[10], and the presence within this region of key beta cell loci such as *INS/IGF2* and *CDKN1C*. The average sequencing depth was 22X, and DNA methylation measurements were obtained from 29,675 CpG dinucleotides (Supplementary Data 3).

**Similarities in methylation patterns**. The overall methylation profile among the insulinomas (Fig. 1a, red line) is highly correlated (bivariate Pearson correlation coefficient = 0.934; $p = 1.93E-61$) to that of beta cells (blue line), compatible with the notion that insulinomas and beta cells share a common cellular origin.

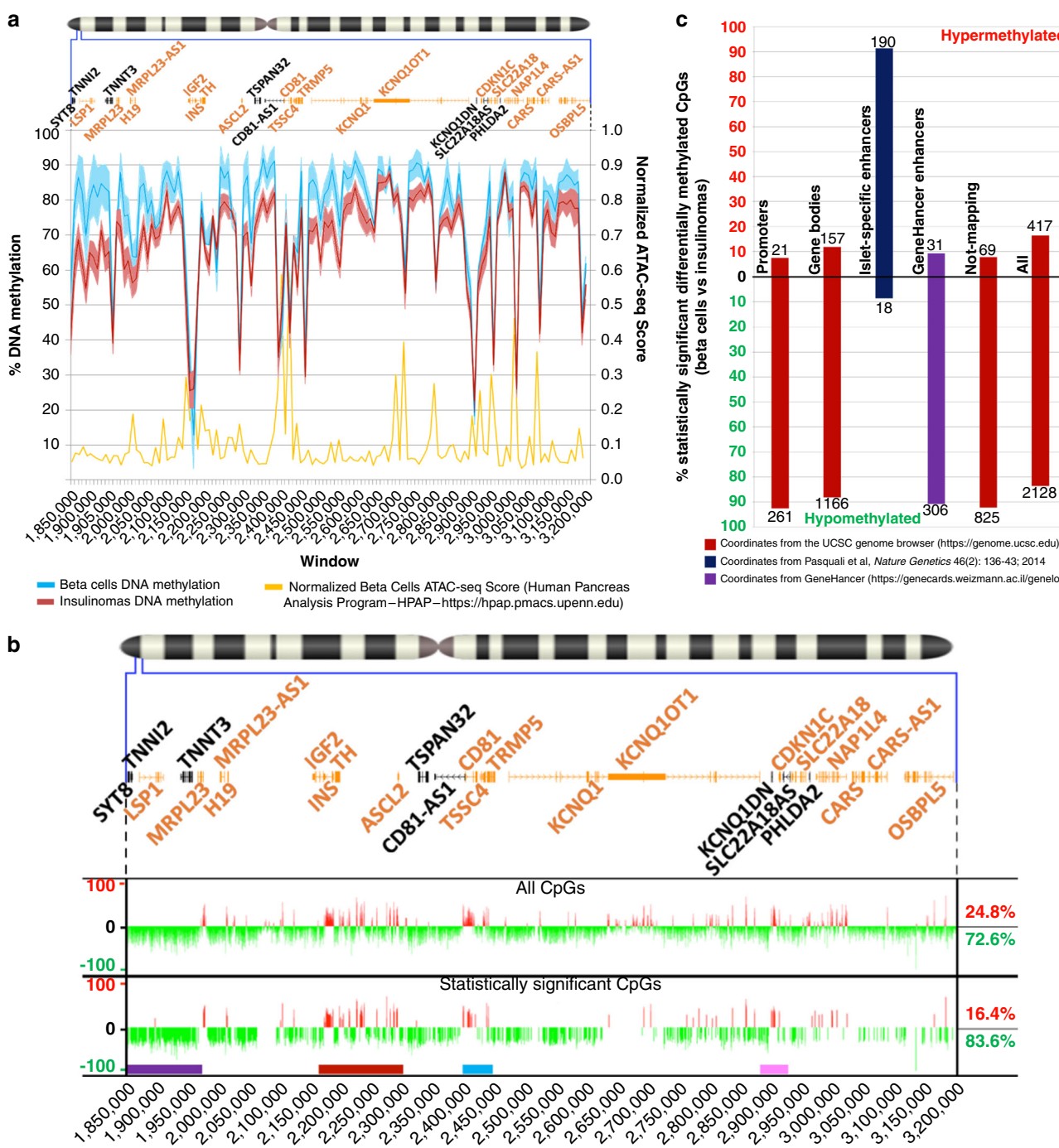

Beta cells DNA methylation
Insulinomas DNA methylation
Normalized Beta Cells ATAC-seq Score (Human Pancreas Analysis Program—HPAP—https://hpap.pmacs.upenn.edu)

Coordinates from the UCSC genome browser (https://genome.ucsc.edu)
Coordinates from Pasquali et al, *Nature Genetics* 46(2): 136-43; 2014
Coordinates from GeneHancer (https://genecards.weizmann.ac.il/geneloc)

Interestingly, and as expected[10], the regions of lower methylation in both beta cells and insulinomas overlap with key beta cell genes, such as *INS/IGF2* and *CDKN1C*. We overlaid the methylation profiles of insulinomas and beta cells onto open chromatin regions in purified human beta cells, as defined by ATAC-seq analysis from the Human Pancreas Analysis Program (HPAP) database[14] (yellow track). This demonstrated a close inverse correlation between open chromatin peaks as assessed by ATAC-seq and regions of hypomethylation (bivariate Pearson correlation coefficients: beta cells $= -0.593 - p = 3.65\text{E}{-}14$; insulinomas $= -0.519 - p = 1.13\text{E}{-}13$) (Fig. 1a and Supplementary Data 4). These latter results are expected, and independently support the accuracy of the methylation sequencing data.

**Differences in methylation patterns**. Despite their similarities, insulinomas clearly differ from beta cells in that they are relatively hypomethylated across the majority of the 11p15.5-p15.4 target sub-region, as visualized by the lower profile of the red line compared to the blue line in Fig. 1a. This difference is supported in statistical terms by the range of values in the blue and red shading, indicating the 5th and 95th percentile confidence intervals. Cluster analysis using the hclust algorithm (Supplementary Fig. 3) visually suggests that that beta cells and insulinomas are distinct. However, dimensional as well as component analyses (Supplementary Fig. 4) failed to support the generation of independent clusters among insulinomas. Visual distribution of insulinomas within the heatmap in Supplementary Fig. 3A

**Fig. 1 Insulinoma and beta-cell DNA differential methylation profiles. a** DNA methylation profiles. (Top) The Chromosome 11 ideogram with a magnification of the 11p15.5-p15.4 target sub-region. In the upper open box, gene placeholders are displayed as per the "UCSC Genes" track (dense display). Black gene names refer to genes that are not expressed and orange names to genes that are expressed in insulinomas or beta cells[10]. (Middle) In the graph, the *X* axis represents a breakdown of the 1.35 Mbp 11p15.5-p15.4 target sub-region in 135 windows of 10 kbp each. The left *Y* axis shows the percent DNA methylation and the right *Y* axis, the normalized ATAC-seq score[14] (Supplementary Data 4). The light blue and red lines show the average DNA methylation distribution across all CpG dinucleotides of each window, with 95% confidence interval shading, for beta cells and insulinomas, respectively. The yellow line shows the open chromatin peaks from four unrelated human beta cells over the same region[14]. **b** Differential DNA methylation tracks. At the top is the chromosome 11 ideogram with a magnification of the 11p15.5-p15.4 target sub-region as described in **a**. Tracks were generated using beta-cell samples as reference, compared to insulinomas. (Top track) All measured CpG dinucleotides. (Bottom track) Statistically significant differentially methylated CpG dinucleotides (beta-binomial distribution of read counts with dispersion shrinkage with the "DMRcate" R package – FDR < 0.005). Four regions were identified with either extensive uninterrupted hypomethylation or prevalent hypermethylation (Supplementary Fig. 8 and Supplementary Data 3 and 5). These are labeled by colored rectangles below the bottom track. Rectangle colors, telomerically (left) to centromerically (right), are purple (coordinates: chr11:1,850,000–1,970,000), dark red (chr11:2,160,000–2,300,000), light blue (chr11:2,396,000–2,440,000), and pink (chr11:2,870,000–2,921,000). **c** Sub-categorization of the statistically significant differentially methylated CpG dinucleotides by genomic element. The *X* axis shows the genomic elements: promoters, gene bodies, islet-specific enhancers, GeneHancer enhancers, regions not mapping to any of the genomic elements considered, and the whole-target region. The *Y* axis reports the percent of significantly hyper-/hypomethylated CpG dinucleotides. Below and above each bar, the numbers of hypo- and hypermethylated CpG dinucleotides, respectively, are shown (Supplementary Data 7 and 8).

suggested they might be sorted into three groups with decreasing degrees of similarity. This prompted us to further assess the existence of differential DNA methylation signatures between each insulinoma group and the beta cells (Supplementary Fig. 5 and Supplementary Data 3). It was noted that the insulinomas in Group 3 (Supplementary Fig. 3) included samples with a very limited degree of similarity with each other or other insulinomas, likely reflecting two unusual insulinoma subtypes (Ins_31, Ins_33) and poor DNA quality in sample Ins_32, which shows the lowest number of measurable CpG dinucleotides among the 19 insulinomas tested. Removal of these samples or other samples with chromosomal loss within the 11p15.5-p15.4 target sub-region (i.e., Ins_23 and Ins_25) did not significantly alter the grouping of the remaining insulinomas and beta cells. In agreement with the cluster analysis, which suggested more commonality than difference among the insulinomas, the annotation track of the differential methylation across all CpG dinucleotides comparing all insulinomas with beta cells (Fig. 1b, Supplementary Fig. 3B, and Supplementary Data 3) showed a remarkable degree of similarity to the tracks for each putative insulinoma group, regardless of grouping method. We further tested potential clustering through a second round of analysis using only CpG dinucleotides with the top 10% variance across samples (Supplementary Fig. 6). The small differences in group memberships and differential methylation confirmed the lack of significant clusters in the data set. These findings support considering all insulinomas as a single group for statistical purposes. Going forward, all insulinoma and beta-cell data sets were compared based on their representing normal vs. abnormal biology.

We worried that a comparison of pure FACS-sorted beta cells to insulinomas that contain cells other than beta cells (e.g., endothelial cells, fibroblasts) may complicate our comparison. The striking similarity between insulinomas and beta cells (Fig. 1c) provides reassurance that if there is contamination by non-beta cell types in the insulinomas, it is not quantitatively important.

Differential methylation analysis comparing all insulinomas to beta cells revealed 2545 differentially methylated CpG dinucleotides in insulinomas (FDR < 0.005), visualized in the bottom annotation track of Fig. 1b. Interestingly, while consistent enough across samples to be strongly significant, the degree of hypo- or hypermethylation of the statistically significant differentially methylated CpG dinucleotides was strongly associated with the group signatures (Supplementary Fig. 7) (chi-squared = 180.41, df =1, $p = 2.2E-16$). Similarly, the distribution of the statistically significant differentially methylated and the signature CpG dinucleotides over the 11p15.5-p15.4 target sub-region is strongly

correlated (bivariate Pearson correlation coefficient = 0.611, $p = 2.2E-16$) (Supplementary Fig. 7). This suggests a particularly significant role for the specific CpG dinucleotides in Fig. 1b in determining differences in the insulinoma phenotype vs. normal beta cells and warranted additional experimental analyses.

**Multiple regions of hypo/hypermethylation in insulinomas.** The differential methylation annotation track in Fig. 1b shows broad areas of hypomethylation in insulinomas vs. beta cells, interrupted by multiple peaks of relative hypermethylation (the red peaks in Fig. 1b). Four regions stood out, most of which are functionally relevant. These four areas carry the highest number of hypo- or hypermethylated CpG dinucleotides (Supplementary Fig. 8), giving rise to some of the differentially methylated regions (DMRs) with the strongest statistical significance (maximum FDR of the statistically significant differentially methylated CpG dinucleotides for each DMR < 0.005) (Supplementary Fig. 8 and Supplementary Data 6). These regions are indicated by the purple, dark red, blue, and pink bars below the bottom track of Fig. 1b. In functional terms, the purple region (chr11:1,850,000–1,970,000) overlaps with the longest uninterrupted and strongly hypomethylated region, and also overlaps with the *SYT8/TNNI2 locus* involved in looping, and potently transactivating the *INS* promoter[11]. The dark red region (chr11:2,160,000–2,300,000) overlaps the *INS/IGF2 locus* and the adjacent centromeric gene desert. The light blue region (chr11:2,396,000–2,440,000) was selected because it represents the second most hypermethylated region. The pink region (chr11:2,870,000–2,921,000) overlaps with *CDKN1C*, the cell cycle inhibitor gene whose loss of function is principally responsible for beta cell proliferation in BWS[7], in FoCHI[8], and contributes to proliferation in insulinoma[10].

At a high level, these apparently insulinoma-specific, recurrent, and multiple regions of differential methylation suggest a pattern of global dysregulation of chromatin architecture in the 11p15.5-p15.4 target sub-region, implying disorganized looping and inappropriately targeted silencing of specific loci. Therefore, we next examined the distribution of the 2545 statistically significant differentially methylated CpG dinucleotides (Fig. 1b, bottom track), assigning them to promoters (defined as 1500 bp upstream and 500 downstream the TSS of each gene isoform), gene bodies, and enhancers across the target sub-region (Fig. 1c). To define enhancers, we drew data from two sources, the GeneHancer database, which integrates 434,000 reported enhancers from four different genome-wide databases, including ENCODE, Ensembl, FANTOM, and VISTA[15] (note that GeneHancer does not contain data from human islets or beta cells), and the human islet regulatory network (islet regulome) of Pasquali et al.[16], which

scored all islet-specific active and inactive enhancers by a combination of FAIRE- and ChIP-Seq. Through counting the number of both significantly hypo- and hypermethylated CpG dinucleotides mapping to each of these genomic elements, a clear trend emerged: promoters, gene bodies, and regions not mapping to promoters, genes or enhancers (red bars) were highly hypomethylated in insulinomas as compared to beta cells (~90% of CpGs) (red bars in Fig. 1c, Supplementary Data 7, 8). In marked contrast, islet-specific enhancers showed the exact opposite profile: they were heavily hypermethylated in insulinomas (~90% of CpGs) (dark blue bars). Notably, this did not apply to enhancers from non-islet cell types (purple bar). These findings strongly suggest that the differential DNA methylation profile in insulinomas targets specific enhancers that affect beta-cell specification, with potential effects on chromatin conformation, as well as both local and long-range transcriptional regulation.

**Promoter methylation does not explain abnormal gene expression.** We compared promoter methylation patterns to differential expression data for the 11p15.5-p15.4 target sub-region from our prior study comparing RNA-Seq from 25 insulinomas vs. 22 sets of FACS-purified human beta cells[10]. Of the 33 genes in the 11p15.5-p15.4 target sub-region, 28 displayed: (a) either both differential promoter DNA methylation *and* differential gene expression; or, (b) either differential promoter DNA methylation *or* differential gene expression. Using the methylation index[17], we graphed expression log-fold change against methylation log-fold change for the 28 genes (Fig. 2a and Supplementary Data 9). The findings were unimpressive: only six genes (*LSP1, H19, TH, KCNQ1, SLC22A18, OSBPL5*) showed significantly opposite methylation and expression changes; five genes (*MRPL23-AS1, IGF2, INS, INS-IGF2, CD81*) showed opposite methylation and expression changes without significant alteration in gene expression; four genes had concordant methylation and expression changes (*ASCL2, TSSC4, TRMP5, CARS-AS1*); six genes (*MRPL23, IGF2-AS, KCNQ1OT1, CDKN1C, NAP1L4, CARS*) had expression data but no methylation data; seven genes, had methylation data but no expression data available (*SYT8, TNNI2, TNNT3, TSPAN32, KCNQ1DN, SLC22A18AS, PHLDA2*). In addition, when conducting the same analysis, individually, on the six insulinomas that underwent both RNA-seq and targeted bisulfite DNA sequencing, the differential methylation and expression trends reported above were confirmed (Supplementary Fig. 9 and Supplementary Data 10). Collectively, these findings make the point that promoter methylation status has little to do with the expression of genes in the 11p15.5-p15.4 target sub-region. This point is further underscored by the observation that the entire target sub-region is overwhelmingly hypomethylated (Fig. 2a). Conversely, the significantly opposite methylation and expression changes exemplified by *LSP1, H19, TH, KCNQ1, SLC22A18, OSBPL5* are thus more likely random and unrelated to insulinoma pathogenesis.

**Chromatin, promoter, enhancer, and dyadic signatures.** To investigate whether the putative promoters of the 28 genes within the 11p15.5-p15.4 target sub-region (Fig. 2b) serve as promoters and/or as enhancers, we used the Roadmap Epigenomics Project database[18], which, through the combination of five histone modification marks across 111 reference epigenomes representative of all major human cell lineages, provides maps of regions carrying histone coding signatures suggestive of promoters or enhancers, or signatures that have a bivalent profile, referred to as "dyadic"[18]. This revealed that fewer than half of the putative promoters (11/28) actually contain histone mark promoter signatures. Instead, 24 of the 28 genes carried enhancer

($n = 21$) and/or dyadic ($n = 10$) histone mark signatures. Thus, the enhancer and dyadic enrichment signature are remarkably prominent across the entire 11p15.5-p15.4 target sub-region (Fig. 2c and Supplementary Data 11). Collectively, these findings suggest that the broadly abnormal differential methylation profile of the 11p15.5-p15.4 target sub-region among insulinomas likely affects enhancers and their networks related to beta-cell phenotype and proliferation; they also suggest that abnormal chromatin looping and chromatin conformation occurs in insulinomas.

**Abnormal methylation targets beta-cell TF binding sites.** To explore whether abnormal methylation in insulinomas might lead to altered chromatin conformation, we turned to ReMap2018[19], a curated database of 2829 publicly available ChIP-Seq data sets covering 485 transcriptional regulators, including transcription factors (TFs), transcriptional co-activators, and chromatin-remodeling factors, such as histone-modifying enzymes. We first generated a baseline of transcriptional regulator binding sites mapping to at least one of the >30,000 CpG dinucleotides in the 11p15.5-p15.4 target sub-region, using a standard enrichment score approach that calculates the proportion of nucleotides of each binding site covered by CpG dinucleotides. We next generated a similar inventory of transcriptional regulator binding sites mapping to at least one of the 2545 statistically significant differentially methylated CpGs (Fig. 3a and Supplementary Data 12). We restricted the list of relevant transcriptional regulators to those expressed in beta cells and insulinomas defined by our human beta-cell and insulinoma RNA-seq data set[10]. For each binding site, we also calculated its differential methylation status and graphed the cumulative methylation change for each transcriptional regulator against its enrichment rank change, the latter calculated by comparing the ranks for significant differentially methylated CpG dinucleotides vs. the baseline (Fig. 3c). Remarkably, using the 5th and 95th percentile for methylation and rank change as cutoffs, we identified four strongly over-enriched and hypermethylated transcriptional regulator binding sites (PDX1, NKX2-2, NKX3-1, NR5A2). Two additional transcriptional regulator binding sites were found to be strongly over-enriched and hypomethylated (NFATC1, STAT5A), and one (TFDP1) was found under-enriched and hypermethylated. Of these, PDX1 and NKX2-2 are essential beta-cell transcription factors[20] and NFATC1 has been shown to promote *INS* expression[21].

We next mapped the transcriptional regulator binding sites showing strong hypo-/hypermethylation and under-/over-enrichment (red and green peaks), strong hypo-/hypermethylation alone (orange and light green peaks), and under-/over-enrichment alone (gray peaks) (Fig. 4a and Supplementary Data 13). Notably, the two principal regions carrying the majority of the mapped sites precisely overlapped with the previously defined hypomethylated purple (chr11:1,850,000–1,970,000) and hypermethylated dark red (chr11:2,160,000–2,300,000) regions in Fig. 1b. A scattering of other less striking peaks was identified across the remaining target regions, one of which overlapped the hypermethylated light blue region (chr11:2,396,000–2,440,000). Perhaps most importantly, the hypermethylated dark red region bears the vast majority of strongly over-enriched and hypermethylated transcriptional regulator binding sites (Fig. 4b). These observations suggest that binding sites for key canonical, beta cell-specific transcription factors, such as PDX1 and NKX2-2, at the *INS/IGF2 locus*[22] are hypermethylated and poorly accessible to their normal transcriptional regulatory machinery in insulinomas. We confirmed the inability of PDX1 to bind to its normal binding sites using qPCR-ChIP analysis (Supplementary Fig. 10 and Supplementary Data 14). Collectively, these unanticipated

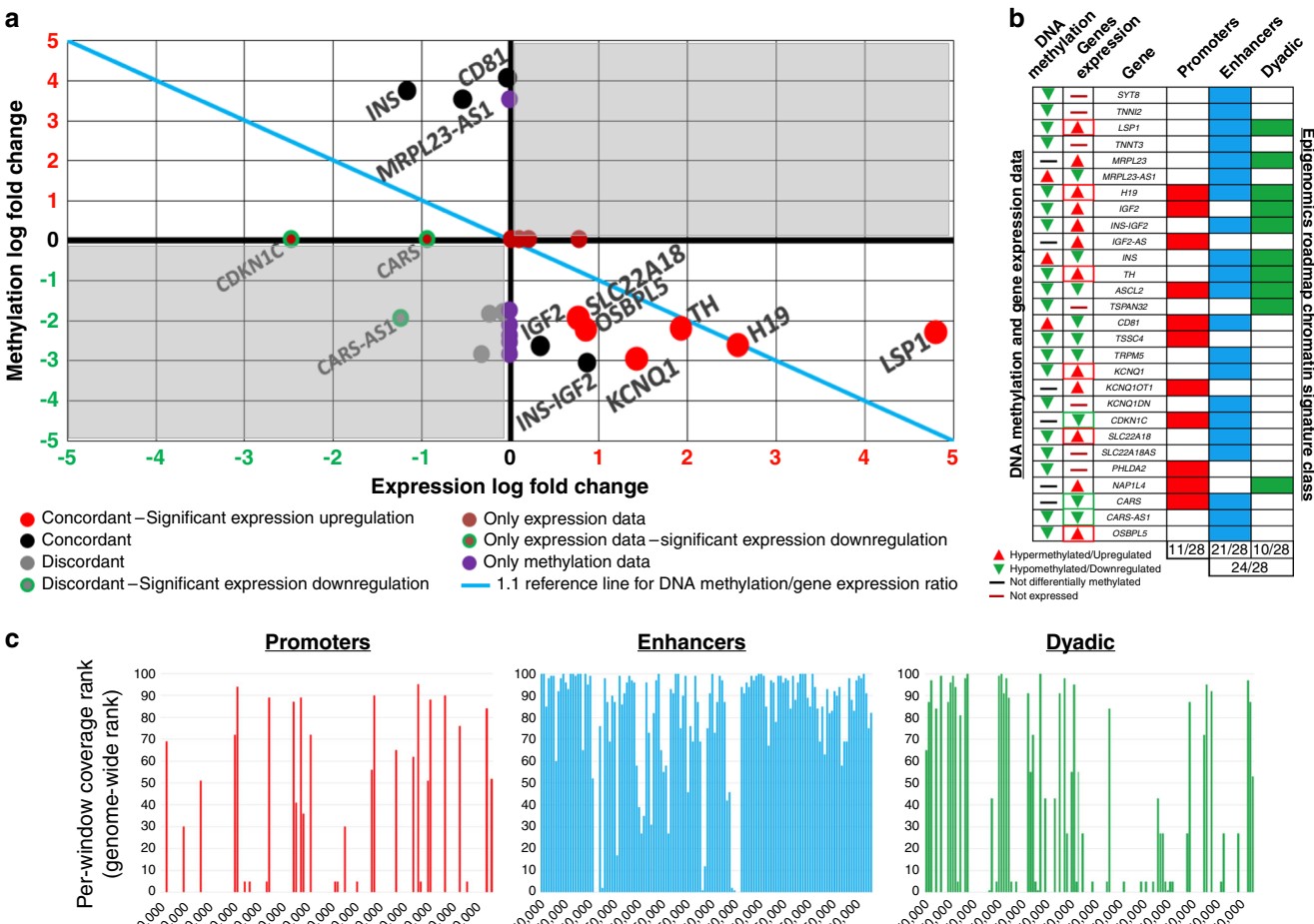

**Fig. 2 Promoter DNA methylation versus gene expression. a** Poor correlation between promoter methylation and gene expression. The *X* axis shows the gene expression log-fold change as reported previously[10]. The *Y* axis reports the DNA methylation log-fold change between insulinomas and beta cells, calculated as per Du et al.[17]. The top right and bottom left corners of the graph are shaded in gray, to represent "discordant" regions of promoter DNA methylation and gene expression. Since it is expected that promoter hypermethylation would favor gene silencing, and promoter hypomethylation would favor gene expression, these quadrants highlight genes that do not follow this axiom (see Supplementary Data 9 for details). **b** Chromatin signature of the promoters in the target region. The rows display the 28 genes in the target region for which either DNA methylation or gene expression[10] data are available. The two columns to the left of the gene names display the promoter DNA methylation status and the expression status of the genes indicated. Red arrowheads indicate statistically significant upregulation and green arrowheads downregulation. For DNA methylation, all data are from statistically significant differentially methylated CpGs. For gene expression, arrowhead red and green boxing represent statistical significance. The last three columns score, as per the Roadmap Epigenomics Project[18], the presence of promoter, enhancer, or dyadic chromatin signatures in the promoter region of each gene. **c** Relative abundance of promoters/enhancers/dyads chromatin signatures in the 11p15.5-p15.4 target sub-region. The *Y* axis reports the genome-wide rank of the coverage of each window by sequences with promoters/enhancers/dyads chromatin signatures. A strong enrichment signal can be detected for enhancers (mean rank = 82, scored windows = 131) as compared to both promoters (mean rank = 52, scored windows = 30) and dyads (mean rank = 53, scored windows = 63). See Supplementary Data 11 for details. Note that the 11p15.5-p15.4 target sub-region is heavily enriched for enhancer signatures.

events suggest that alternate transcriptional regulatory mechanisms must drive insulin gene over expression in insulinomas, and also possibly affect other genes transcribed from the 11p15.5-p15.4 target sub-region.

To explore this possibility, we next cross-referenced the differential methylation of transcriptional regulator binding sites in the 11p15.5-p15.4 target sub-region with our insulinoma-beta cell RNA-seq differential expression data set[10] (Fig. 5). We focused on the hypomethylated purple (chr11:1,850,000–1,970,000) and hypermethylated dark red (chr11:2,160,000–2,300,000) regions that carried most of the strong hypo- and hypermethylated and/or under- or over-enriched transcriptional regulator binding sites. Within the hypermethylated dark red region containing the *INS/IGF2* locus, *PDX1* and *NKX3-1* showed both a significant

reduction in expression and strong hypermethylation/over-enrichment of their binding sites (Fig. 5a, right and Supplementary Data 15). This would provide an opportunity for other transcriptional regulators to control the expression from the *INS/IGF2 locus* in insulinomas. The hypomethylated purple (chr11:1,850,000–1,970,000) region was of particular interest, for it bears the CTCF binding site within the *SYT8/TNNI2* locus which has been reported by Xu et al.[12] to make contact with the *INS* promoter in normal human islets. Within this purple region, the binding site for only one transcriptional regulator, NFATC1, showed both a significant increase in expression and strong hypomethylation/over-enrichment (Fig. 5a, left and Supplementary Data 15). This suggests that NFATC1 may serve as an alternate driver of *INS/IGF2* expression in insulinomas, possibly

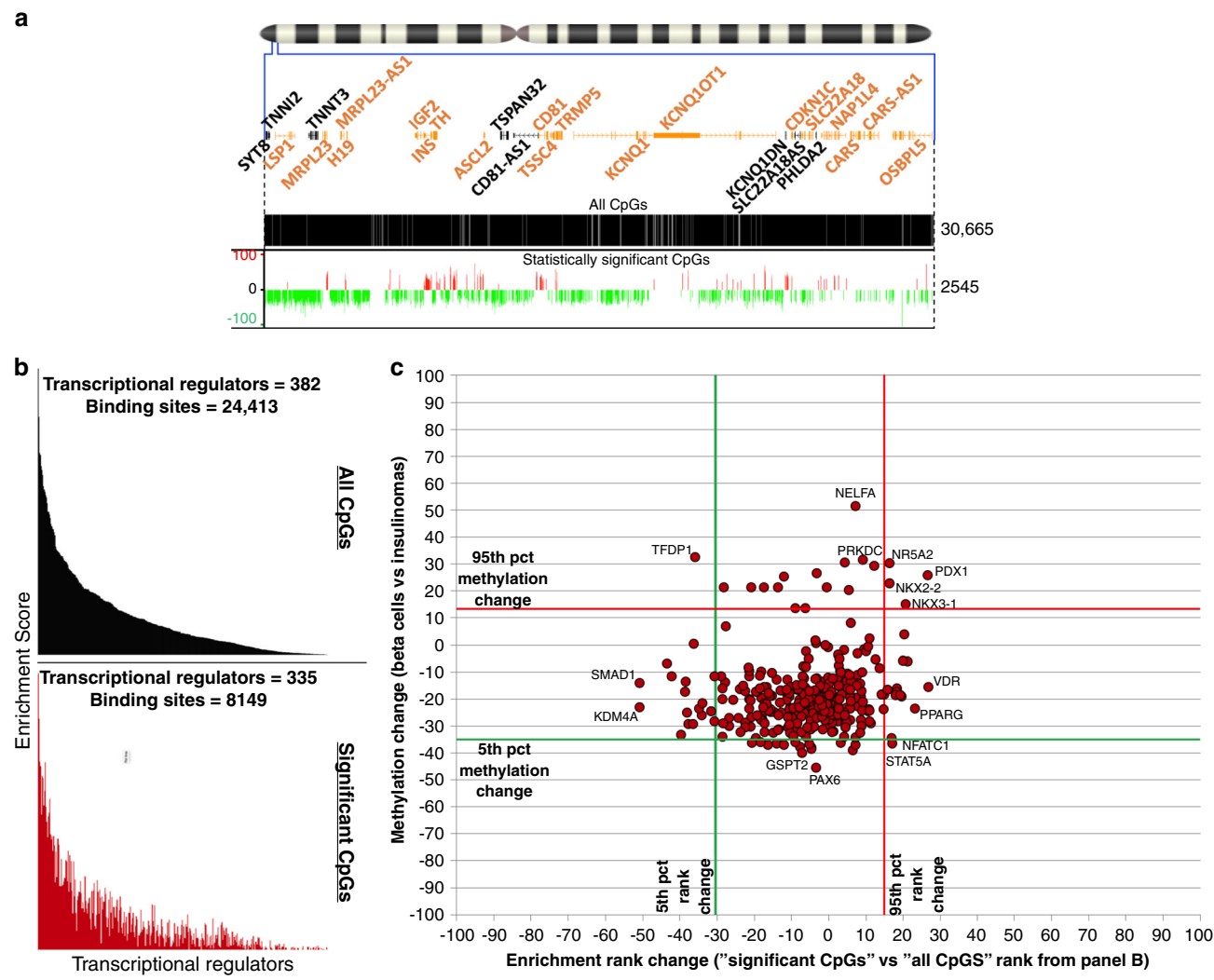

**Fig. 3 Transcriptional regulator analysis for differentially methylated regions. a** The chromosome 11 ideogram with a magnification of the 11p15.5-p15.4 target sub-region is as described in Fig. 1a. The black track displays the distribution of all 30,665 CpG dinucleotides populating the target region. The red-green track on the bottom is the same as the bottom track in Fig. 1b, showing 2545 differentially methylated CpG dinucleotides in insulinomas vs. beta cells. **b** Distribution of transcriptional regulator binding sites. For both panels, the X axis represents the transcriptional regulator binding sites from the ReMap2018 database[19] carrying either CpG dinucleotides that are located in the target region (top—black graph), or CpG dinucleotides with significant differential methylation (bottom—dark red graph). Transcriptional regulator names are shown in Supplementary Data 12. Only transcriptional regulators that were expressed in either beta cells or insulinomas were considered for this analysis. The Y axis of each graph displays the enrichment score calculated as the fraction of nucleotides of each transcriptional regulator binding site covered by CpG dinucleotides. For both graphs, transcriptional regulator binding sites are displayed using the same rank as for the "All CpGs". This panel illustrates the differential enrichment of transcriptional regulators between the two tracks (Supplementary Data 12). **c** Enrichment rank and methylation change for the transcriptional regulator binding sites. The X axis reports the enrichment rank change when comparing the baseline "All CpGs" and the "Significant CpGs" graphs of **b** normalized to the number of transcriptional regulators of the two curves. The Y axis reports, for each transcriptional regulator binding site, the cumulative DNA methylation, calculated by averaging the DNA methylation within each binding site, which in turn was obtained by averaging the significant differentially methylated CpG dinucleotides (**a** bottom track). Red and green lines identify 5th (green) and 95th (red) percentile for rank (vertical) and methylation (horizontal) change. Remarkably, key pancreatic transcription factors such as PDX1 and NKX2-2 appear both over-enriched and hypermethylated (see Supplementary Data 13 for details).

facilitated by looping from the *SYT8/TNNI2 locus*. Interestingly, NFATC1 is overexpressed by a log2-fold factor of 3 in insulinomas as compared to normal beta cells (Supplementary Data 16).

To further explore transcriptional control in this region, we mapped all binding sites in the 11p15.5-p15.4 target sub-region for the three transcriptional regulators (PDX1, NKX3-1, NFATC1) identified in Fig. 5a against their hypo- or hyper-methylation status (Fig. 5b and Supplementary Data 17). Remarkably, PDX1 and NFATC1 had the highest and best matching percentage of differentially methylated transcriptional regulator binding sites {14/30 sites (47%) for both}, compared to

NKX3-1 {18/43 (42%)}. PDX1 and NFATC1 differentially methylated binding sites were also all hyper- or all hypomethylated, respectively, while NKX3-1 showed a more mixed profile (4 hypo- and 14 hypermethylated sites). In addition, while no differentially methylated PDX1 sites could be found mapping to regions hosting NFATC1 sites or vice versa, NKX3-1 sites mostly mapped within PDX1-rich regions and, in one case, within an NFATC1-rich region. Finally, 12 out of 14 hypermethylated NKX3-1 sites had sequences overlapping with PDX1 sites as curated by ReMap2018[19] (Supplementary Data 17). Taken together, these observations most likely reflect the incidental

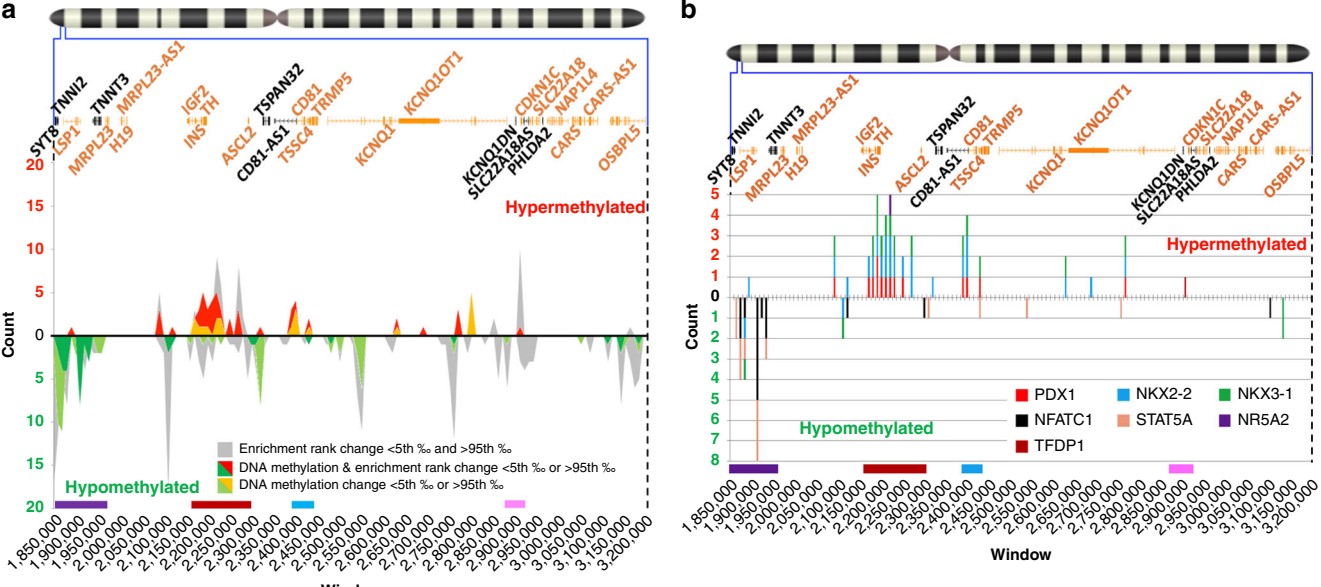

**Fig. 4 Distribution of enriched and/or differentially methylated transcriptional regulator binding sites.** For both panels, the chromosome 11 ideogram with a magnification of the target 11p15.5-p15.4 target sub-region is as described in Fig. 1a. **a** Distribution of the enriched/differentially methylated transcriptional regulator binding sites across the target region. The gray, red, yellow, and green peaks display the locations of enriched/differentially methylated transcriptional regulator binding sites. The X axis shows a breakdown of the 1.35 Mbp target sub-region in 135 windows of 10 kbp each. The Y axis reports the count of transcriptional regulators per 10 kbp window. Above and below the "0" black line, the locations and number of hypo- and hypermethylated transcriptional regulator binding sites are displayed as peaks. Strong peaks of transcriptional regulator binding site hypo- and hypermethylation can be seen overlapping purple, dark red, and light blue regions defined in Fig. 1b. **b** Distribution of differentially enriched/methylated (<5th ‰ and >95th ‰) transcriptional regulator binding sites. The X axis is as in **a**. breaks down the 1.35 Mbp target sub-region into 135 windows of 10 kbp each. The Y axis reports the count of transcriptional regulators per window. Above and below the "0" black line, the numbers of hyper- and hypomethylated transcriptional regulator binding sites are shown in the stacked histogram bars. Most of the enriched and differentially methylated transcriptional regulator binding sites reside in the purple and dark red regions.

inclusion of NKX3-1 in this analysis and suggest that PDX1 and NFATC1 binding sites as the key targets of the differential methylation of insulinomas.

We also explored the light blue (chr11:2,396,000–2,440,000) and pink (chr11:2,870,000–2,921,000) regions (Supplementary Fig. 11 and Supplementary Data 15). While more limited in the number of hypo- or hypermethylated and/or under-/over-enriched transcriptional regulators, for the most part, the hypermethylated light blue region mimicked the hypermethylated dark red region profile (Supplementary Fig. 11). The hypermethylated pink region did not generate a consistent profile.

From the foregoing findings, a coherent picture emerges, suggesting that the transcriptional control of the *INS/IGF2* locus in insulinomas, may, at least in part, switch from local or proximal PDX1-based transactivation to a more distal transcriptional regulation, driven by looping of the hypomethylated purple region containing the CTCF binding site within the *SYT8/TNNI2 locus* and/or other loci to the *INS* promoter, as first reported by Xu et al.[12]. In addition, the $log_2$ 3× increase in *NFATC1* expression in insulinomas vs. beta cells, together with the hypomethylation of the multiple NFATC1 binding sites in the *SYT8/TNNI2 region*, suggests that insulin gene transactivation in insulinoma may employ transcriptional signals derived from NFATC1.

**Enhancer binding sites and methylation profiles.** Enhancers participate in determining chromatin structure to support the transcriptional needs of a given cell type[23]. They do so in collaboration with transcriptional regulators, the binding sites for which are often abundant in enhancers[24]. The findings in Figs. 1c, 2c, and 4a support such a functional role in beta cells mediated by

islet-specific enhancers, while these same sites are hypermethylated in insulinomas. To more deeply explore whether the differential methylation profile of enhancers in insulinomas might affect the chromatin structure of the 11p15.5-p15.4 target sub-region, we next mapped all of the transcriptional regulator binding sites to the genomic elements of Fig. 1c, and calculated their density ratio (Fig. 6a and Supplementary Data 18). As expected, the highest transcriptional regulator binding site density was found, in descending order, in enhancers, followed by promoters, and finally gene bodies and non-mapping regions, with islet-specific enhancers being strikingly and disproportionately hypermethylated. The islet-specific enhancers collectively cover only an approximate 49.9 kbp, substantially less than half of the range covered by GeneHancer enhancers (~113 kbp) and promoters (~115 kbp), and an order of magnitude smaller than regions containing gene bodies (~700 kbp) and non-mapping regions (~490 kbp). Remarkably the islet-specific enhancers include 927 transcriptional regulator binding sites (175 hypo- and 752 hypermethylated) for some 251 transcriptional regulators out of the 335 mapping within the entire 1.35 Mbp 11p15.5-p15.4 target sub-region (Fig. 6a). The likely relevance of these islet-specific enhancers to the differential methylation profile of insulinomas is further underscored by their distribution along the 11p15.5-p15.4 target sub-region compared to the distribution of all transcriptional regulator binding sites (Fig. 6b and Supplementary Data 18): clusters of islet-specific enhancers appear precisely in regions carrying strong peaks of hypermethylated transcriptional regulator binding sites. In contrast, the non-islet-specific GeneHancer enhancers are scattered across the 11p15.5-p15.4 target sub-region, overlapping only minimally with islet-specific enhancers. Together, these findings further support the

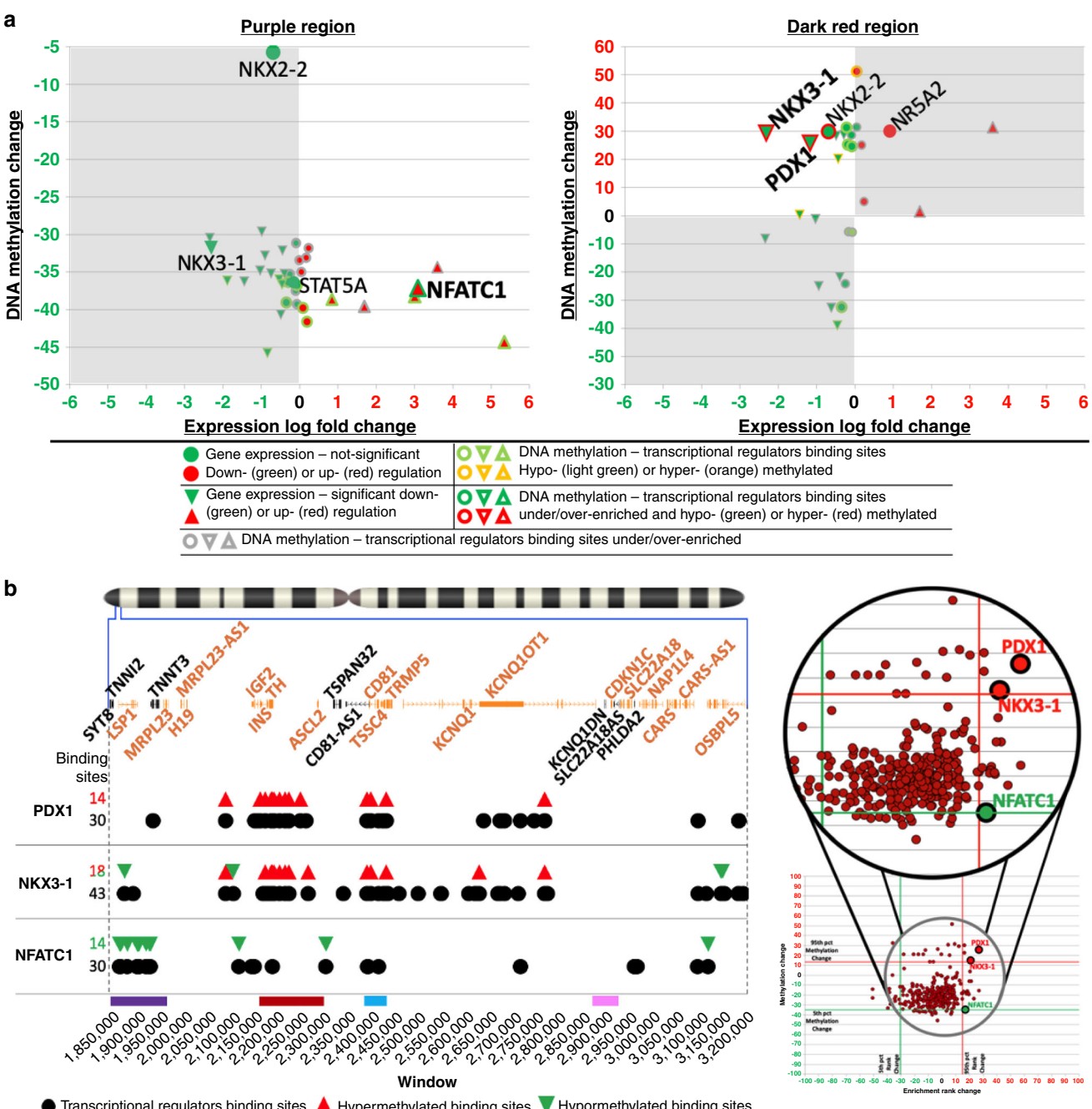

**Fig. 5 Identifying the key transcriptional regulator binding sites in the purple and dark red regions. a** Comparison of transcriptional regulator binding site DNA methylation vs. transcriptional regulator expression. For both graphs, the X axis reports the expression fold change[10] and the Y axis the cumulative methylation change between insulinomas and beta cells. Following the same approach as in Fig. 2a, for each transcriptional regulator, reduced binding site availability (hypermethylation) should be accompanied by a reduction in its expression. Conversely, increase binding site availability (hypomethylation) should overlap with increased expression. Thus, the portions of both graphs with discordant methylation and expression directions are shaded in gray. The transcriptional regulator binding sites that are differentially enriched/methylated (<5th ‰ and >95th ‰) are shown (see Supplementary Data 15 for details). Only transcriptional regulators NFATC1, in the purple region, and PDX1 and NKX3-1, in the dark red region, show concordant enrichment/methylation (<5th ‰ or >95th ‰) and significant expression dysregulation. **b** The chromosome 11 ideogram with a magnification of the 11p15.5-p15.4 target sub-region is as described in Fig. 1a. Comparative distribution of all vs. the differentially methylated transcriptional regulator binding sites identified in **a**. The X axis provides a reference for the position of the transcriptional regulator binding sites across the target region. The Y axis represents the list of the transcriptional regulators from the dark red (PDX1, NKX3-1) and purple (NFATC1) regions. Each row illustrates the mapping of all binding sites for that transcriptional regulator (dark black circles) vs. differentially methylated (upward red triangle = hypermethylated; downward green triangle = hypomethylated) binding sites. PDX1 binding sites in the red region solely carry hypermethylated CpG dinucleotides and are presumably less accessible to the binding of the PDX1 transcription factor. NKX3-1 sites reveal a mixed profile that correlates only moderately with its significantly downregulated expression. NFATC1 binding sites solely carry hypomethylated CpG dinucleotides and are thus presumably available for NFaT binding (Supplementary Data 17). The panel on the right is adapted from Fig. 3c and highlights the methylation/enrichment of PDX1, NKX3-1, and NFATC1 binding sites in Fig. 3c.

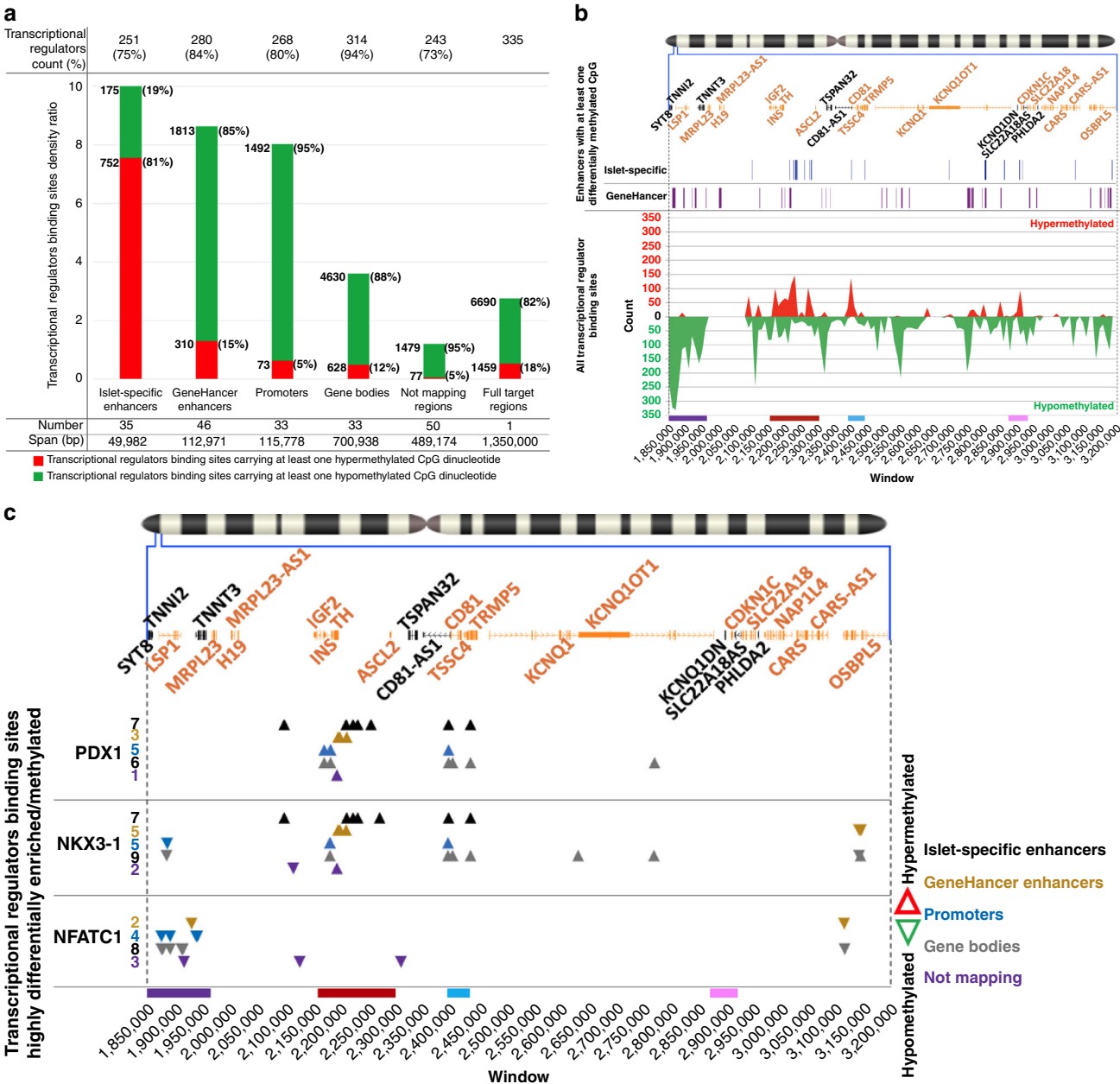

**Fig. 6 Mapping transcriptional regulator binding sites by genomic elements. a** Histogram of transcriptional regulators by genomic elements. The *X* axis lists the same genomic elements of the target region in Fig. 1c. The *Y* axis reports the ratio of the cumulative length of the binding sites mapping in each region over the cumulative length of the region itself. The red and green coloring of the histogram refers to the proportion of hypo- and hypermethylated sites. Each red and green portion of each histogram bar is labeled with the number of hypo- and hypermethylated binding sites. Above the histogram, for each genomic element, the count and percentage are shown for the number of transcriptional regulators per genomic element. Below the histogram, the number of genomic elements and their total span, in base pairs, is also shown (Supplementary Data 18). **b** Top panel: The chromosome 11 ideogram with a magnification of the 11p15.5-p15.4 target sub-region as described in Fig. 1a. Middle panel: Tracks showing the distribution of 35 islet-specific enhancers (blue track) and 46 generic, GeneHancer enhancers (purple track). Bottom panel: The *X* axis represents a breakdown of the 1.35 Mbp target sub-region in 135 windows of 10 kbp each. The *Y* axis shows the count, per window, of all differentially methylated transcriptional regulator binding sites. Above and below the "0" black line, the count of hypo- and hypermethylated transcriptional regulator binding sites are reported as peaks. Note the lack of overlap between islet-specific and GeneHancer enhancer and, conversely, the strong overlap between islet-specific enhancers and dark red and purple regions shown in the bottom of the figure (see Supplementary Data 18 for details). **c** Map of binding sites for the transcriptional regulators in Fig. 5 by the genomic elements in **a**. The *X* axis provides a reference for the position of the transcriptional regulator binding site across the target region. The *Y* axis shows the transcriptional regulators for the dark red (PDX1, NKX3-1) and purple (NFATC1) regions. Each row provides the mapping of the significant differentially methylated binding sites by genomic elements (see Supplementary Data 18 for details).

notion that that the islet-specific enhancers have a critical role in shaping normal chromatin structure in the 11p15.5-p15.4 target sub-region in normal beta cells, and in remodeling this structure in insulinomas.

We next mapped the transcriptional regulator binding sites from the defined hypomethylated purple (chr11:1,850,000–1,970,000) and hypermethylated dark red (chr11:2,160,000–2,300,000) regions in Fig. 5 to all genomic elements (enhancers, promoters, gene bodies, and non-mapping regions) throughout the entire 11p15.5-p15.4 target sub-region. As shown in Fig. 6c, only hypermethylated PDX1 and NKX3-1 binding sites in insulinomas mapped to islet-specific enhancers (7/14), and the majority of these (4/7) were near the *INS/IGF2 locus*. All but one site were shared by the two transcriptional regulators, again suggesting: (1) incidental inclusion of NKX3-1; and, (2) more importantly, the disruption of PDX1 binding to (Supplementary Fig. 10), and altered regulation of expression from, the *INS* locus in insulinomas. In contrast, NFATC1 binding sites distributed broadly among promoters, enhancers, gene bodies, and non-mapping regions (Supplementary Data 18). These latter observations suggest that NFaT transcription factors may respond to non-islet-specific signals that participate in modifying chromatin structure in the 11p15.5-p15.4 target sub-region, thereby affecting the expression pattern of the *INS/IGF2 locus* in insulinomas.

**Modeling INS regulation in human insulinomas.** EndoC-βH1 cells are a T-antigen- and Tert-immortalized proliferating human embryonic pancreatic beta-cell line used to model human beta cells[25]. Jian and Felsenfeld have performed genome-wide Circularized Chromosome Conformation Capture (4C)-Seq on EndoC-βH1 cells using an anchor primer (anchor region) positioned over and centromeric to the *INS* promoter[13]. This study identified physical contacts between the anchor region and sites across the genome. To further understand chromatin interactions in the 11p15.5-p15.4 target sub-region in insulinomas, we integrated their peak calls for DNA loop frequency from EndoC-βH1 cells into our beta cell-insulinoma data sets. Specifically, we overlaid the DNA loop frequency onto the transcriptional regulator binding site chart in Fig. 4a, positioning the anchor over, and centromeric to, the *INS* promoter, as had Jian and Felsenfeld. As shown in Fig. 7a, a striking overlap appeared between the highest DNA loop frequency in the 4C-Seq data set and the hypermethylated and over-enriched binding sites in Fig. 4a, specifically including the PDX1 binding site (Supplementary Data 19). These direct experimental data strongly support the concept that the PDX1 binding site serves as an anchor for a PDX1-based enhancer network in normal beta cells. More importantly, in the insulinoma context, the observation that these PDX1 binding sites are hypermethylated and less accessible in insulinomas than in beta cells, and the failure of PDX1 to bind to these same sites in insulinoma cells (Supplementary Fig. 10), strongly suggests that alternate, non-PDX1 mechanisms, including sites in the hypomethylated purple region, mediate *INS*, and possibly *INS-IGF2*, expression in insulinomas. This hypothesis is further supported by recent work by Lawlor and Stitzel[26] who performed Hi-C sequencing on EndoC-βH1 cells. The results show DNA looping occurring within the hypermethylated dark red region that extends from the *INS/IGF2 locus*, centromerically, over the PDX1 sites of this region. Moreover, additional DNA loops bridge the telomeric side of the hypermethylated dark red region with the *SYT8/TNNI2* locus, underscoring the common occurrence of these interactions in this cell line.

## Discussion

This study reveals multiple important insights into beta cell biology and insulinoma pathogenesis. First, it represents a large

and comprehensive next-generation deep methylome sequencing data set and bioinformatic analysis of the imprinted 11p15.5-p15.4 sub-region in normal adult human beta cells. Second, it provides a complete data set of CpG methylation status in human insulinoma. Third, it provides a comprehensive bioinformatic comparison of this key imprinted region in normal human beta cells and human insulinomas, a region that is also implicated in beta cell hyperplasia in BWS and FoCHI. Fourth, in contrast to the large variety of variants associated with insulinoma[10], we observe that all insulinomas share an abnormal, surprisingly uniform, methylome signature within the beta-cell phenotype-determining 11p15.5-p15.4 sub-region, complementing an equally uniform clinical and pathologic phenotype. And fifth, we provide a model to explain the abnormal proliferation and insulin overproduction characteristic of human insulinoma.

Methylation patterns in the 11p15.5-p15.4 target sub-region previously have been explored in human insulinomas, but limited to two imprinted control regions (ICRs) and two imprinted ancillary sequences. In 2009, Dejeux et al.[27], studying 11 insulinomas, described apparent hypermethylation in 40 CpGs across the *H19/IGF2* ICR (a.k.a. ICR1) and two of its ancillary imprinted sequences. In 2016, Bhatti et al.[28] described similar methylation patterns in 9 insulinomas occurring in children across 3 CpG dinucleotides of the *H19/IGF2* ICR; in addition, hypomethylation was detected across 3 CpG dinucleotides of the kvDMR1 ICR (a.k.a. ICR2). Both studies, however, lacked relevant normal cell types as controls, surveyed a very limited number of CpG dinucleotides, and scored differential methylation by assuming baseline hemi-methylation at each imprinted region. Here, we extend this coverage from a few CpGs to all 30,665 CpGs in the 11p15.5-p15.4 target sub-region, and also provide an appropriate tissue-specific control, the normal beta cell. In contrast to these smaller studies, we find that the predominant methylation abnormality in this region is hypomethylation, punctuated by focal areas of intense and reproducible hypermethylation. Parenthetically, the two ICRs in this region, the H19/IGF2 ICR (chr11: 2,018,812–2,024,740) and KvDMR1 (chr11: 2,719,948–2,722-259), each displayed approximately equal percent methylation in beta cells (49.5% and 41.9%, respectively) and insulinomas (47.9% and 36.5%, respectively).

Previously, we described deep CpG methylome sequencing of the same 11p15.5-p15.4 target sub-region, but were limited to only two sets of FACS-sorted beta cells and ten insulinomas[10]. That study suggested widespread methylation abnormalities in this region in insulinomas, that likely affected the expression of key genes in the *INS/IGF2 locus*, the *CDNK1C*, and others. However, this prior study was underpowered to assess the statistical significance and functional implications of the methylation changes observed. Accordingly, here we doubled the numbers of beta-cell and insulinoma methylomes, to gain a clearer picture of the spectrum of the 11p15.5-p15.4 target sub-region methylation changes, their reproducibility, and their functional implications. Further, we have integrated our methylome-seq findings with other large data sets including purified beta-cell gene expression[10], open chromatin (ATAC-Seq) analysis, enhancer and promoter determination, transcriptional regulator binding sites, and 4C-Seq chromatin structure data. Collectively, these yield a surprisingly uniform model of gene expression control in the imprinted region in human insulinoma.

The model is summarized in Fig. 7b. A key unanticipated finding is that the 19 insulinomas differ from the five beta-cell data sets with regard to the methylation status of the *INS* region, where the key beta-cell transcription factor, PDX1, binds, recruits other enhancer members, and transactivates *INS* gene expression. In normal beta cells, this PDX1 binding region is lightly methylated, and accessible by PDX1. In contrast, in insulinomas, the

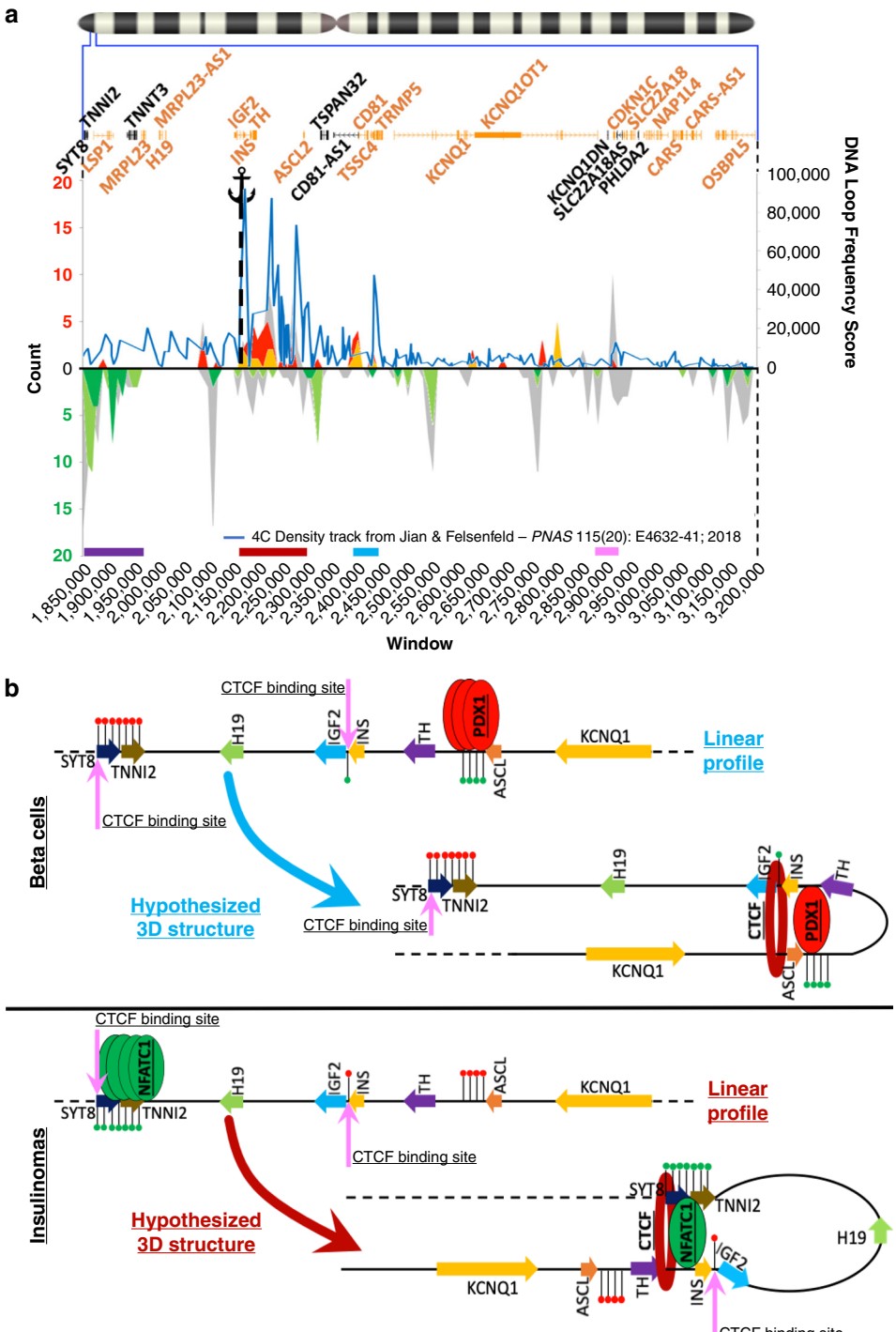

PDX1 binding region is hypermethylated and therefore less accessible to PDX1. Moreover, the expression of *PDX1* itself is reduced in insulinomas. In isolation, these events would predict that *INS* gene expression should be low or absent in insulinomas. Yet, the opposite is true: *INS* gene expression is markedly elevated in insulinomas compared to all other normal cells; indeed insulin hypersecretion is a *sine qua non* of insulinoma diagnosis. This begs the question, "if PDX1 and other canonical beta-cell transcription factors cannot access and drive *INS* gene expression in insulinoma, what *does* drive *INS* gene expression?". Using elegant 4C-Seq approaches in the EndoC-βH1 human beta-cell line, Jian and Felsenfeld[13] showed that the anchor makes frequent contacts with a region centromeric to the *INS/IGF2* locus that overlaps

with our hypermethylated dark red region and contains the majority of the PDX1 binding sites, which are hypermethylated in insulinomas. In addition, clear but less frequent contacts were detected between the anchor and the *SYT8/TNNI2 locus*, findings recently confirmed by Hi-C sequencing in EndoC-βH1 cells[26].

The *SYT8/TNNI2* locus is included in the hypomethylated purple region which carries the majority of the NFATC1 binding sites. Xu et al.[12] have also shown that normal human islets treated with high glucose form highly frequent enhancer-like DNA loops between the *SYT8/TNNI2 locus* and the *INS* promoter, aided by a CTCF binding site located within the *SYT8* gene sequence. These loops become more frequent as high glucose stimulates *INS* expression[13]. These findings suggest that in beta cells and EndoC-

**Fig. 7 Predicted effect of altered methylation profile on 11p15 chromatin structure. a** Comparison of beta cell and insulinoma transcriptional regulator binding sites to *INS* promoter-based chromatin loops derived from EndoC-BH1 beta cells[11]. The top panel containing the chromosome 11 ideogram with a magnification of the 11p15.5-p15.4 target sub-region is as described in Fig. 1a. The lower panel superimposes the 4C frequency track from EndoC-βH1 cells (blue line) on the hypermethylated and hypomethylated transcriptional regulator binding sites shown as peaks in Fig. 4a. The anchor symbol illustrates the anchor primer used in the 4C experiments, positioned over and centromerically to the *INS* promoter[11]. The *X* axis represents a breakdown of the 1.35 Mbp target sub-region in 135 windows of 10 kbp each. The left *Y* axis reports the count of transcriptional regulators per 10 kbp window. The right *Y* axis represents the 4C DNA looping score. This is not summarized per window but instead refers to specific coordinates (Supplementary Data 19). **b** A cartoon summarizing events predicted in normal beta cells (top panel) and insulinomas (bottom panel). In beta cells, under normal conditions, *INS* transcription is driven by PDX1 (and related transcription factors and enhancer networks) that bind the unmethylated or lightly methylated PDX1 binding sites. CTCF-cohesin loops ensure PDX1 contact with the PDX1 binding sites. In contrast, in insulinomas, the PDX1 binding sites are hypermethylated, *PDX1* expression is downregulated, and altered methylation rearranges CTCF-cohesin boundaries so that telomeric regions near the *SYT8/TNNI2* locus are brought into proximity to the *INS* promoter region, and transactivate *INS*, as proposed by Xu et al.[12]. Finally, the abundance of NFaT transcription factors and their unmethylated binding sites in the *SYT8/TNNI2 locus* suggest that NFaTs may be involved in enhancing the activity of this/these new loop(s). In both panels, the red lollipops indicate hypermethylation, the green lollipops hypomethylation, the pink arrows CTCF binding sites derived from the UCSC genome browser. Both panels contain a linear version of the region as well as a cartoon displaying the proposed 3D structure. Adapted from Xu et al.[12] and Jian and Felsenfeld[13].

---

βH1 cells, expression from the *INS/IGF2 locus* is regulated by PDX1 binding within the dark red region, and that additional regulation is provided by the purple region that, as shown in normal human islets, upon treatment with high glucose, gives rise to an enhancer loop that transactivates *INS* expression[12], effectively replacing PDX1 in this role (Fig. 7b). However, in contrast to the Xu, Jian, and Felsenfeld findings in normal beta cells, in the insulinoma model, this transient loop is formed constitutively, reflecting the loss of 11p15.5-p15.4 3D chromatin structure, presumably related to its broad hypomethylation. Formal chromatin structural assays will be required to test this model.

The hypermethylated dark red region, centromeric to the *INS/IGF2 locus* contains a number of narrowly spaced, islet-specific enhancer sites and a remarkable number of transcriptional regulator binding sites (Fig. 7b), notably those for PDX1. NKX3-1 binding sites in this region generally overlap with PDX1 binding sites as curated by ReMap2018 (Figs. 5b and 7b), and may also be sites for another important beta-cell transcription factor, NKX2-2 (Fig. 4b). Because of this, and since *NKX3-1* expression is downregulated in insulinomas[10], we suspect that the NKX3-1 binding sites captured in our analysis are a bioinformatic anomaly, but may suggest additional layers of complexity to the model in Fig. 7b. Also, since NKX2-2 and NKX3-1 binding sites are similar, since NKX2-2 is critical for the beta-cell phenotype, and is abundant in both normal beta cells and insulinomas, it is possible that the predicted NKX3-1 binding sites may also serve as NKX2-2 binding sites.

NFaTs are transcription factors that are present in the cytoplasm of beta cells under basal conditions, tethered to scaffold proteins of the 14-3-3 class. They are released from 14-3-3's by the phosphatase calcineurin, and traffic to the nucleus where they are able to engage response elements in promoters of genes related to beta-cell proliferation (such as cyclins, cdks, *FOXM1*) and beta-cell differentiation (such as insulin, glucokinase, Glut 2, chromogranin A, IAPP)[21,29,30]. Keller et al.[31] have demonstrated that over expression of constitutively active mouse Nfatc1 and Nfatc2 enhance insulin secretion from mouse islets. These events in mouse beta cells translate to normal human beta cells: the ability of NFaTs to remain in the nucleus and perform these actions is maintained by beta cell regenerative drugs that inhibit the kinase, DYRK1A[2–6]. Thus, it was of great interest to observe that expression of *NFATC1* is increased by a factor of three-fold in insulinomas and that other NFaT family members are altered[10], and that its cognate response elements at the telomeric end of 11p15.5-p15.4 target sub-region are hypomethylated, and therefore presumably accessible to the NFaT family. Whether NFaTs preferentially bind in insulinomas—as compared to beta

cells—to distal 11p15.5-p.15.4 sites, as suggested by the model in Fig. 7b, and whether this alters *INS* expression and regional 3D chromatin structure, as the model may predict, also will be of great interest to decipher in future studies. We also note that since NFATC1 and NFATC2 binding motifs are similar, it is possible that the NFATC1 binding sites may also serve as binding sites for NFATC2 as proposed by Keller et al.[31].

Despite their common clinical and pathological features and RNA-seq profiles[10], insulinomas contain variants in a broad range of epigenetic modifying enzymes, exemplified by Trithorax, Polycomb, and related members such as *MEN1, KDM6A, MLL3/KMT2C, YY1, KDM5B*, and *SMARCC1*. As an example, inactivating mutations in *MEN1*, an H3K4me3 methylase and a canonical member of the Trithorax family, are common in sporadic and familial insulinomas. Inactivating *MEN1* mutations are an unequivocal cause of germline insulinoma in mouse and man[9,10,32,33]. It remains uncertain precisely whether or how mutations in Trithorax, Polycomb, and related genes are related to altered 11p15 methylation, or whether the epigenetic variants and dysregulated 11p15.5-p15.4 methylation are both necessary but independent events. For example, it is conceivable that the methylation abnormalities either cause or result from loss of positioning of normal insulators in the course of tumorigenesis, possibly a result of mutations in Trithorax, Polycomb, or related genes. Conversely, the precise cause of the broad regional demethylation and local hypermethylation in the 11p15.5-p15.4 target sub-region is also unknown. It is also uncertain whether it is acquired in adulthood (when most insulinomas are detected) or early in prenatal of postnatal life. Further studies will be required to address these complex issues.

Insulinoma, FoCHI, and the BWS are considered to be independent syndromes. However, they all share abnormalities in the 11p15 imprinted region, all are associated with the focal expansion of beta-cell clusters within the pancreas, all are associated with markedly reduced expression of *CDKN1C* (encoding the cell cycle inhibitor p57$^{KIP}$), and in all, beta-cell proliferation is accelerated. These observations raise a question as to whether these are truly distinct syndromes, whether there may be etiologic overlap among these three syndromes, or whether differing 11p15 methylation or structural abnormalities may converge on similar syndromic endpoints. As an example, it is easily conceivable that focal 11p15 methylation abnormalities may result not only in insulinoma but also FoCHI. Clarifying these issues will require deep methylome sequencing of surgically resected tissue samples from multiple infants with FoCHI and BWS, a challenging goal because of the rarity of these syndromes.

These studies focus on control of both insulin gene expression as well as proliferation in insulinoma, and we find clues to the abnormal regulation of *INS* gene expression (Fig. 7b) as well as of proliferation (e.g., loss of *CDKN1C* expression) in insulinoma. However, it remains unclear whether the 11p15 events are specific to insulinomas, or whether they might apply to all PNETs, most of which do not produce or secrete excessive amounts of insulin. For example, is an insulinoma an "insulinoma", and not "non-functioning PNET", because of the 11p15 methylome abnormalities, or do these abnormalities exist in non-insulinoma PNETs? To our knowledge, there are no deep methylome 11p15 sequencing data available from non-insulinoma PNETs that might help to clarify this issue.

Overall, these studies reveal a consistent, important, and previously unrecognized pattern of 11p15.5-p15.4 methylation in human insulinomas. They suggest a model that may explain why *INS* is overexpressed in insulin-producing PNETs. We suggest that future studies be undertaken to explore 11p15 methylation status in non-insulinoma PNETs, FoCHI, and BWS. Further, 3D chromatin structural studies, such as 4C, Hi-C, and CTCF ChIP-seg studies that might unequivocally demonstrate abnormal chromatin organization in insulinomas are required to validate the model in Fig. 7b. Defining what is abnormal will first require defining the normal 11p15 chromatin looping pattern in normal beta cells. This will require miniaturization of 3D chromatin structural assays and their deployment on normal human beta cells. Defining "normal" itself is a challenge because there is no readily available supply of purified human beta cells, nor a perfect human beta-cell line. It also remains to be determined whether the Trithorax/Polycomb abnormalities in insulinomas are a cause, result, or are unrelated to the methylation abnormalities within the 11p15.5-p15.4 target sub-region. Finally, defining the role of NFaT family member signaling in controlling the 3D conformation of the 11p15.5-p15.4 target sub-region and *INS* gene expression will be important next steps in future studies.

## Methods

**Tissue samples**. Purified human beta cells were isolated from seven human cadaveric islets donors provided by the NIH/NIDDK-supported Integrated Islet Distribution Program (IIDP) (https://iidp.coh.org/overview.aspx), the University of Chicago and the Alberta Diabetes Institute. Informed consent was obtained by the Organ Procurement Organization (OPO), and all donor information was de-identified in accord with Institutional Review Board procedures at The Icahn School of Medicine at Mount Sinai (ISMMS), The Alberta Diabetes Institute and the University of Chicago. Demographic information is provided in Supplementary Data 1. Beta cells were labeled using adenoviral labeling with ZsGreen, chemical labeling with Newport green, or insulin labeling, followed by flow cytometric sorting[4,10,34,35]. Two sets of beta cells bisulfite sequenced from four islet donors (one sample derived from a single islet donor and one sample comprises DNA pooled from three donors) were reported previously[10], and beta-cell DNA from three new donors are included here as described in Supplementary Data 1 and Supplementary Fig. 1. Nineteen human insulinomas were collected from subjects who provided informed written consent and were deposited in the ISMMS Biorepository. Patient samples were de-identified through the Biorepository and Pathology Core at the ISMMS (IRB-HSM-00145) or the providing institutions. Clinical details of the patients with insulinoma are provided in Supplementary Data 2. Ten were included in a prior study[10] and nine additional insulinomas were added for the current study. In the original study[10], since we were seeking previously unrecognized SNVs/MNVs/Indels in non-canonical insulinoma genes, insulinomas derived from patients known to be members of *MEN1* kindreds were intentionally excluded a priori. In the current study, one subject (Ins_27) had a germline *MEN1* mutation, a Trp183Stop change (Supplementary Data 2).

**Targeted DNA methylation analysis**. The 11p15.5-p15.4 Target Sub-Region: The target region selected for this analysis extends from coordinate 1,850,000–3,200,000 between bands p15.5-p15.4 of chromosome 11 (GRCh37/hg19 Assembly). Bisulfite DNA Sequencing: Two sets of DNA (300 ng) from five sorted adult beta-cell samples were utilized. Three-hundred ng of DNA were obtained from four samples (Beta_1, Beta_3, Beta_4, Beta_5), and three other sorted beta cell DNA preparations were pooled to achieve the 300 ng minimum for bisulfite sequencing (Beta_2). Three-hundred ng DNA was prepared from 19 insulinomas, of which six had also previously been analyzed by RNA-Seq (Ins_11, Ins_13, Ins_18, Ins_22, Ins_23,

Ins_24)[10]. Ten had been reported previously and nine were new samples (Supplementary Data 2). Probes were designed to capture the 11p15.5-p15.4 target sub-region according to the SeqCap Epi Enrichment System (NimbleGen, Roche Sequencing & Life Sciences, Indianapolis, IN, USA) for hybridization-based targeted enrichment of bisulfite-treated DNA. Pre-capture libraries were prepared using the Kapa Hyper Prep kit, PCR-free version (Roche Sequencing & Life Sciences). The 24 samples were barcoded and twelve-plex-sequenced in three runs (other unrelated samples were also sequenced)[10]. Genomic DNA was sonicated using a Covaris S220 (Covaris, Woburn, MA) to an approximate length of 180–220 bp. End-repair and A-tailing were performed followed by ligation of methylated SeqCap Epi indexed adaptors. Products were cleaned using Agencourt AMPure XP beads (Beckman Coulter, Indianapolis, IN). Bisulfite conversion was carried using the Zymo EZ DNA Lightning kit (Zymo Research, Irvine, CA), followed by ligation-mediated PCR amplification with HiFi HotSart Uracil + polymerase (Roche Sequencing & Life Sciences). Multiplex hybridization was performed on bisulfite converted libraries to the custom SeqCap Epi Choice probe pool. Hybridized products were purified with Capture Beads and PCR amplified to generate the final libraries for sequencing. Final yields were quantified in a Qubit 2.0 Fluorometer (Life Technologies, Grand Island, NY), and library quality was assessed on a DNA1000 Bioanalyzer chip (Agilent Technologies, Santa Clara, CA). Post-capture multiplexed libraries were normalized, pooled, clustered on a V2 paired-end read flow cell, and sequenced for 150 cycles on an Illumina MiSeq (Illumina, San Diego, CA). The primary processing of sequencing images was done using Illumina's Real Time Analysis (RTA) software. Sequencing was performed at the Epigenomics Core Facility at The Weill-Cornell College of Medicine. Insulinoma Sequencing Consistency: To assess internal consistency, one of the insulinoma samples, Ins_11, was sequenced in each of the three twelve-plex runs, with samples blinded to the epigenomics sequencing core. When analyzed in a correlation matrix with all other insulinomas using the hclust algorithm with the ward method ("corrplot" R package), the Ins_11 replicates are those that showed the highest correlation among all insulinoma samples (bivariate Pearson correlation coefficients > 0.859, $p = 2.2E−16$) supporting the reliability of the multiplexed sequencing approach used for this experiment (Supplementary Fig. 12). Pooled beta-cell sample methylation profile: to assess the reproducibility of sequencing and native consistency among normal beta cell populations from the seven different donors, the correlation between the pooled and individual beta cells samples was tested by bivariate Pearson correlation. This correlation was strong (bivariate Pearson correlation coefficient = 0.959 – $p = 8.84E−63$). In addition, clustering analysis across all samples scored the five beta-cell data sets as those with the smallest distance (Supplementary Fig. 3A).

**DNA methylation pipeline**. CASAVA 1.8.2 software was used to de-multiplex samples, generate raw reads and respective quality scores. Raw data were quality filtered, adapter trimmed, and reads aligned to the bisulfite converted reference human genome (GRCh37/hg19—whole-genome alignment) and the methylation context for each cytosine determined. BSMAP was used to compute the percent methylation scores and average conversion rates. The average conversion rates obtained ranged from 99.53 to 99.79%. All CpG dinucleotide methylation calls generated by the alignment pipeline were filtered for a minimum sequencing depth of 5× for each CpG. The full list of all calls for all samples across the target region is provided in Supplementary Data 3 in percent methylation format.

**Comparative and statistical analyses**. Comparative analyses were conducted using data downloaded from online databases listed and briefly described in each sub-section below. All statistical analyses employed were performed using R, version 3.5.3.

DNA methylation profiles were generated by (1) dividing the 1.35 Mbp 11p15.5-p15.4 target sub-region into 135 windows of 10 Kbp each; and, (2) averaging the DNA methylation of all CpG dinucleotides in each window. For ATAC-Seq, we downloaded the ATAC-Seq peaks from four purified human beta cells samples from the HPAP database (https://hpap.pmacs.upenn.edu/explore/download). Since scores are provided by HPAP as non-normalized peaks, we first normalized, for each sample, the individual peaks to the total signal across the 11p15.5-p15.4 target sub-region. We then selected, for each 10 kbp window, for each sample, the highest ATAC-Seq score, and averaged it across the four samples before plotting. We selected this approach to prevent dilution of the ATAC-Seq signal across windows resulting from the intrinsic nature of the ATAC-Seq output, which often provides a high number of low-intensity peaks surrounding a limited number of high-intensity peaks (see Supplementary Data 4 for details).

For DNA methylation clustering and dimensional and component analyses, we used the heatmap2 function of the "gplots" R package to generate the methylation heatmap on our data set of all, and top variable, CpG dinucleotide methylation percentage calls across our sample set. We then analyzed the data dimensionality by using scree, variance and principal component analysis plots from the "stats" R package. By using the hclust algorithm from the same "stats" R package we finally verified that no meaningful clusters could be generated and instead visually distributed insulinomas in three groups by using the heatmap for all CpGs as reference. For the signature analysis, given the limited and varying number of samples per each of the group that was arbitrarily generated, we opted for using a simple multi-step procedure to determine each groups signature as of: (1) within each

insulinoma group, discarding those CpG dinucleotides that had a standard deviation of the differential methylation between each insulinoma and the average of all beta cells above the 95th percentile for its distribution; (2) among the remaining CpG dinucleotides, selecting those that returned measurable DNA methylation values in each of to the three groups; (3) within each group, for each CpG dinucleotide, calculating the average differential methylation; (4) calculating the standard deviation of the average differential methylation for the three groups; and finally (5) selecting those CpG dinucleotides with standard deviation above the 95th percentile as group signature. We also compared the "signature" CpG dinucleotides with the distribution of the statistically significant differentially methylated CpG dinucleotides across the 135 windows of 10 kbp of the 11p15.5-p15.4 target sub-region by bivariate Pearson correlation (see Supplementary Data 3 for details).

To determine statistically significant differentially methylated CpG dinucleotides, we used the "DMRcate" R package, which employs a beta-binomial distribution of read count with dispersion shrinkage to score the differentially methylated CpG dinucleotides between the five beta cell and 19 insulinoma samples. Given the limited number of samples available, we included as differentially methylated only CpG dinucleotides with an FDR < 0.005 (see Supplementary Data 3 for details). We also made use of the "DMRcate" function for the determination of differentially methylated regions (DMRs) to support the identification of regions of hypo- and hypermethylation in insulinomas (see below). "DMRcate" scores DMRs using a Gaussian kernel bandwidth for smoothed-function estimation of 1000 bp and a scaling factor for bandwidth. Results are filtered by dispersion shrinkage for sequencing data-derived FDRs for individual CpG dinucleotides as DMR constituents. We selected DMRs containing statistically significant differentially methylated CpG dinucleotides with a maximum FDR < 0.005 (see Supplementary Data 6 for details).

DNA methylation tracks were generated by using the "custom track" function of the UCSC Genome Browser (available at: https://genome.ucsc.edu). Identification of regions of hypo- and hypermethylation in insulinomas. We identified four sub-regions of interest by: (1) calculating the number of statistically significant CpG dinucleotides per each of the 135 10 kbp window of the 11p15.5-p15.4 target sub-region; (2) selecting windows containing a number of hypo- or hypermethylated CpG dinucleotides above the 95th percentile of their distribution across the windows and with at least 75% CpG dinucleotides either hypo- or hypermethylated (see Supplementary Fig. 8 and Supplementary Data 5 for details); and (3) using the DMRs scored by "DMRcate" to determine the coordinates of each region. The four regions identified were finally color-coded, telomerically to cenrtomerically, as purple (coordinates: chr11:1,850,000–1,970,000), dark red (coordinates: chr11:2,160,000–2,300,000), light blue (coordinates: chr11:2,396,000–2,440,000) and pink (coordinates: chr11: 2,870,000–2,921,000).

The distribution of statistically significant differentially methylated CpG dinucleotides by genomic element was determined as follows. Coordinates for promoters and genes were downloaded from the UCSC Genome Browser. For promoters, we considered 1500 bp upstream and 500 bp downstream the transcription start site (TSS) for each gene isoform; for genes, we considered the maximum gene footprint by using the coordinates of the isoforms extending the farthest in both 3′ and 5′ directions. General enhancer coordinates were downloaded from the GeneHancer database (https://genecards.weizmann.ac.il/geneloc/index.shtml), which integrates 434,000 reported enhancers from four different genome-wide databases, including ENCODE, Ensembl, FANTOM, and VISTA[15] (Supplementary Data 7). Islet-specific enhancers coordinates were obtained from the Human Islet Regulatory Network (http://www.isletregulome.org)[16], which scored, among others, all islet-specific active and inactive enhancers derived from a combination of FAIRE- and ChIP-seq (Supplementary Data 7). The distribution of statistically significant differentially methylated CpG dinucleotides across each genomic element was calculated using the "bedtools" suite with the "bedr" package R interface and counting the cumulative number of hypo- and hypermethylated CpG dinucleotides per element.

To compare the differential DNA methylation of each promoter in the 11p15.5-p15.4 target sub-region versus the expression of its cognate gene, we first calculated the methylation index for each promoter of each gene isoform as per Du et al.[17]. Briefly, the methylation index is calculated as:

$$M_i = \log_2\left(\frac{\text{Beta}_i}{1 - \text{Beta}_i}\right)$$

where $M_i$ is the methylation index calculated for the $i$th CpG nucleotide and $\text{Beta}_i$ is the ratio of methylated reads over the total reads for the $i$th CpG dinucleotide. The methylation index provides DNA methylation values in $\log_2$ format which are easily comparable with $\log_2$ fold changes provided by the differential gene expression analysis. For our comparison, we thus determined the methylation index log-fold change between insulinomas and beta cells for each promoter isoforms and then selected those with the best correlation between methylation and expression log-fold changes, i.e., those promoter isoform with ratio methylation over expression log-fold change best approaching 1. We finally compared the methylation vs expression log-fold change trends (Supplementary Data 9).

To obtain data on the chromatin signature of promoters, enhancers, and dyads, we used the Roadmap Epigenomics Database (https://egg2.wustl.edu/roadmap/web_portal/index.html)[18]. Using the combination of five histone modification marks across 111 reference epigenomes representative of all major human lineages, Roadmap Epigenomics provides maps of regions carrying histone coding

signatures which, among others, can be assigned to promoters or enhancers or may have a bivalent profile, referred to as dyadic. We used the Roadmap Epigenomics coordinates to map promoters, enhancers, and dyadic chromatin signatures across the promoters of our gene set. We also graphed promoter, enhancer, and dyadic chromatin signatures from Roadmap Epigenomics for the 11p15.5-p15.4 target sub-region, by calculating the span, percent coverage, and rank of each signature over 135 10 kbp windows (Supplementary Data 11).

To analyze transcriptional regulator binding site enrichment and methylation, we downloaded the list of all transcriptional regulator binding sites from ReMap2018 (http://tagc.univ-mrs.fr/remap/index)[19], a curated database of the binding sites for 485 transcriptional regulators, including transcription factors, transcriptional co-activators, and chromatin-remodeling factors, across 346 cell types, from 2829 publicly available ChIP-seq data sets extracted from the Gene Expression Omnibus (GEO), ArrayExpress (AE) and ENCODE databases, covering 46% of the human genome. We filtered the full list from ReMap2018 for those binding sites in the 11p15.5-p15.4 target sub-region, and for those transcriptional regulators expressed in beta cells and insulinomas, derived from our human beta cell and insulinoma RNA-seq data set[10]. Using the "bedtools" suite and the "bedr" package, we then mapped all 30,665 CpG dinucleotides in the 11p15.5-p15.4 target sub-region onto the binding sites for each transcriptional regulator. We then calculated, for each binding site, the percent coverage by CpG dinucleotides, and summed these for each transcriptional regulator. Transcriptional regulators were then ranked based on their enrichment score. This approach generated a baseline against which we compared the rank obtained by equally mapping and calculating the coverage for the statistically significant differentially methylated CpG dinucleotides. Differential ranks obtained by comparing baseline ranks and ranks for differentially methylated CpG dinucleotides were adjusted for the different number of transcriptional regulators scored by each analysis. The transcriptional regulator binding site cumulative differential methylation was obtained for each binding site by calculating the average differential methylation out of the statistically significantly differentially methylated CpG dinucleotides mapping to each site. Site-specific differential methylation was then averaged across all binding sites for each transcriptional regulator (Supplementary Data 12).

To define differentially methylated transcriptional regulator binding site density per genomic element, we used the "bedtools" suite and the "bedr" package. We mapped all differentially methylated transcriptional regulator binding sites in islet-specific enhancers, GeneHancer enhancers, promoters, gene bodies, and in regions not mapping to the preceding elements. We then calculated the differentially methylated transcriptional regulator binding site density by summing the binding site lengths and dividing it by the cumulative length of each genomic element (Supplementary Data 18).

To assess 4C-Seq DNA loop frequency, Jian and Felsenfeld generously provided 4C-Seq frequency scores from EndoC-βH1 beta cells[13]. The scores for our 11p15.5-p15.4 target sub-region were filtered and graphed across the same region. Note that DNA loop frequency peaks presented herein are not normalized by the distance to the viewpoint (Supplementary Data 19).

**Chromatin immunoprecipitation (ChIP)-qPCR assay**. ChIP was performed using the EZ-ChIP Kit (Millipore) according to the manufacturer's protocol. Three batches of cadaveric human islets (250–300 IEQs) were used per experiment for each PDX1 immunoprecipitation. Insulinoma tissue was chopped into 2-3mm³ pieces and disassociated. Cells were counted using a hemocytometer. Chromatin was prepped from 250,000–300,000 cells according to the EZ-ChIP Kit protocol. The primer sets were designed based on the hypermethylated CpGs in insulinomas and PDX1 ChIP-seq peaks in whole human islets reported by Pasquali et al.[16] and shown in Supplementary Fig. 10 (see Supplementary Data 14 for primer sequences). Immunoprecipitated DNA was quantified using ABI 7500 real-time quantitative PCR detection system (Life Technologies). The anti-PDX1 antiserum (AB2027) was kindly provided by Prof. Christopher Wright at Vanderbilt University and was diluted 1:100 for each ChIP experiment. Data are presented as fold-enrichment of the ChIP signal over the IgG signal.

**Reporting summary**. Further information on research design is available in the Nature Research Reporting Summary linked to this article.

## Data availability

The methylome DNA sequencing data have been deposited in the NIH/NIDDK Diabetes Genotype and Phenotype (dbGaP) database [https://www.ncbi.nlm.nih.gov/projects/gap/cgi-bin/about.html] under the accession code phs001422.v1.p1. The source data underlying Fig. 1, Supplementary Figs. 3, 5, 6, and 7 and Supplementary Data 3 are provided in dbGaP. All the other data supporting the findings of this study are available within the article and its supplementary information files and from the corresponding author upon reasonable request. A reporting summary for this article is available as a Supplementary Information file.

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

## Acknowledgements

We wish to thank Bonnie and Joel Bergstein, and Lonnie and Thomas Schwartz for their constant support of this project. We also thank the NIDDK Human Islet Research Network (HIRN), the NIDDK Integrated Islet Distribution Program, the NIDDK P-30 Einstein-Sinai Diabetes Research Center, The Human Islet and Adenoviral Core at Mount Sinai, The Dean's Flow Cytometry CORE at Mount Sinai, Dr. Piotr Witkowski at the University of Chicago, The Alberta Diabetes Institute, and the Epigenomics Core Facility at The Weill-Cornell College of Medicine. We also thank Drs. Gary Felsenfeld and Xing Jian at the Laboratory of Molecular Biology at the NIDDK for sharing DNA data files, and Dr. Christopher Wright at Vanderbilt University for sharing his PDX1 antiserum. This work was supported by NIH/NIDDK grants R-01 DK116873, P-30 020541, and JDRF Grant 2-SRA-2017 514-S-B, and UC-4 DK098085. R.V.T. and M.S. are supported by the United Kingdom MRC program grants G9825289 and G1000467, National Institute for Health Research (NIHR) Oxford Biomedical Research Centre Programme (R.V.T. and M.S.). R.V.T. is a Wellcome Trust Investigator and NIHR Senior Investigator.

## Author contributions

A.F.S. and L.L. conceived of and initiated the project. E.K., H.W., Y.K., Y.A., A.U., C.A., E.E.S., M.S., Y.L., X.L., F.J., P.W., R.V.T., D.K.S., and L.L. performed laboratory or sequencing work and/or contributed to bioinformatics analysis. M.D., W.I., S.L., G.F.-R., H.S., and R.V.T. helped with tissue acquisition.

## Competing interests

A.U., E.E.S., and H.W. are employees of, and equity holders in, Sema4. The remaining authors declare no competing interests.
