## [Peer Review File · Nature Communications]

Reviewers' comments:

Reviewer #1 (Remarks to the Author):

COMMENTS FOR AUTHOR

This study and a prior study by the same authors (Ref #10) have investigated the genomics and transcriptomics of human Insulinomas. The prior study found abnormal broad DNA hypomethylation in the imprinted chromosome 11p15.5-p15.4 region in insulinomas (n=10) compared to normal beta cells preps (n=2). The work was therefore extended to perform deep DNA methylome analysis of the same region in the current study using 9 additional insulinomas (total n=19), and 3 additional normal beta cells preps (total n=5).

Insulinoma-specific DNA methylation abnormalities in the 11p15 target region when analyzed together with transcription factor binding sites and enhancers predicted altered promoter and enhancer usage in insulinomas (in the 11p15 region). Also, published 3-D structural abnormalities in chromatin looping correlated with hypermethylated PDX1-binding sites at the INS gene in insulinomas indicating non-PDX1-mediated regulation of the INS gene.

Some of the conclusions/interpretations are speculative as mentioned at the end of each paragraph of the Discussion section for future studies, however, the results of this study will be useful to understand the basis of excessive proliferation and insulin expression/secretion by the beta cells in insulinomas, with a potential to inform beta cell regeneration strategies and/or to modulate insulin gene expression.

Some points to strengthen the interpretation and presentation of the work:

1) Clarification about the level of insulin gene expression in human insulinomas: Discussion section page 11 bottom "INS gene expression is markedly ELEVATED in insulinomas compared to most normal cells". Page 8 middle, "Collectively, these surprising events suggest that alternate transcriptional regulatory mechanisms must drive insulin gene OVEREXPRESSION in insulinomas". And page 13 bottom, "Overall, these studies reveal a consistent, important and previously unrecognized pattern of 11p15.5-p15,4 methylation in human insulinomas. They suggest a model that may explain why INS is OVEREXPRESSED in insulin-producing PNETs". However, NO CHANGE in INS expression is seen in Suppl Table-9 and current Fig. 2A (and text page 7 middle), and the same dataset in Ref #10 Fig 2b: Differentially expressed protein-coding genes in insulinomas (n=25) as compared to beta cells (n=22). "insulin mRNA, is EXPRESSED EQUALLY in beta cells and insulinomas". Why are the interpretations different about insulin gene expression between the analysis in the current study and Ref #10? Are there other studies where elevated INS gene expression (at the RNA level) was observed in insulin-producing PNETs (insulinomas) compared to normal beta cells? Those References can be included in the manuscript. Also, if there are any relevant References for the elevated level of NFATC1 gene expression in insulinomas they can be included in the manuscript.

2) Methods, RNA-Seq, relevant to the data presented in the Results section:

2a) How many of the 7 normal beta cells preps were subjected to RNA-Seq (pooled or individual preps)? Which sorting method was used (as per Ref #10) for the 7 beta cells preps: ZsGreen-sorted beta cells, Newport green-sorted beta cells or beta cells from insulin-antibody sorting?

2b) How many of the 19 insulinomas were subjected to RNA-Seq?

“from 19 insulinomas, six had also previously been analyzed by RNA-Seq (Ins_11, Ins_13, Ins_19, Ins_22, Ins_23, Ins_24)”. However, Ins_19 is not listed from previous study, in Suppl Table 2. Does this mean that RNA-Seq data from only 5 insulinoma samples (not 6) was used in the current study? Were these 5 samples compared one to one for gene expression at the 11p15 locus with their respective 11p15 DNA methylation data?

2c) Gene expression data shown in Figure 2 and Figure 4. Not clear whether these Figures compare RNA-Seq data of all 25 insulinoma samples in Ref #10 with the DNA methylation data of 19 insulinoma samples in the current study? Note that among those 25 RNA-Seq insulinoma samples of Ref #10, only 5 have corresponding DNA methylation data. Also, regarding normal beta cells RNA-Seq, are those data from the 22 normal beta cells preps sequenced in Ref #10 or do they correspond to the 7 normal beta cells preps used in the current study?

3) Comparing 3D chromatin structure data of embryonic pancreatic beta cell line (EndoC-βH1 beta cells) with adult beta cells (normal or insulinoma): Are there any differences expected in DNA methylation at the promoter or transcription factor binding sites due to age-related methylation? Is there any known epigenetic difference between fetal/embryonic beta-cell INS gene regulation and adult beta-cell INS gene regulation?

4) Suppl Table 2: Insulin >50 units (in samples Ins_11, Ins_15, Ins_24, and Ins_32) and Insulin <10 units (in samples Ins_12, Ins_13, Ins_16, and Ins_22). Are there any extreme differences in INS or NFATC1 or PDX1 gene expression and 11p15.5-15.4 DNA methylation in these 8 samples with very high and low insulin?

5) Results, first paragraph bottom: Subject with germline MEN1 gene mutation: Is the mutation a VUS or obviously inactivating (stop, frameshift, etc.)?

Reviewer #2 (Remarks to the Author):

This manuscript by Stewart and colleagues is an important follow up work to the Wang et al. 2017 paper. They have now increased the sample size of human insulinomas from ~10 to 38. Additionally they focused their efforts here on the methylation pattern of a critical imprinted region of chromosome 11 containing genes such as INS, IGF2. Interestingly they find that many of the islet specific enhancers are hypermethylated in insulinomas compared to beta cells. Their findings also suggest that insulinomas may possess a different mechanism than beta cells to express insulin. This reviewer finds this point particularly interesting and it should be expanded upon. There are some other conceptual and technical questions and comments that would strengthen this study and enhance its impact to the readers of Nature Communications.

Major comments:

1. It would be interesting to comment on the other differentially expressed genes in insulinomas outside of the 11p15 region now that they have expanded to 38 insulinomas. The previous study reported on ~25 insulinomas. In this study they also found 3 groups of insulinomas. Might this 11p15 region map or predict global gene expression? Directionality/causality would of course be hard to

prove.

2. It possible, it would be compelling to see some data on global genomic methylation on the insulinomas compared to beta cells with this expanded set of insulinomas. Aside from the critical genes in 11p15, how unique or special is it compared to the rest of the genome?
3. The authors state that methylation status surprisingly has weak correlation to gene expression in Figure 2A. However, there could be a threshold effect to the methylation status. The data shown could support a threshold phenomenon. Do the results change if a more absolute level of methylation is used rather than methylation fold change?
4. This study would greatly benefit from either finding out the molecular reason for the hypermethylation patterns in the region of interest. Additionally if any experiments could be done to reverse the methylation status, how this might change gene expression.
5. The authors state an interesting hypothesis that there could be non-PDX1 and NFATC1 dependent transcription of the INS gene for insulinomas. Any positive experimental evidence would be most welcome here. Alternatively, are there other examples in non-insulinoma conditions where NFATC1 could drive INS, such as beta cell failure, dedifferentiation, etc...
6. What is the protein expression of PDX1, NFAT, NKX3 in these insulinomas (especially for first 2)? Helpful to have some idea of protein expression and localization in insulinomas compared to beta cells.

Minor comments:

1. Low quality sequencing of insulinomas (e.g. #32)- if these are thrown out, do the insulinomas still cluster to the 3 groups?
2. Authors should comment on potential limitations with assessing beta cell heterogeneity within the insulinomas on interpreting their results.
3. Figures 1D, 3C and throughout- make clear in the legend and figure that the comparison is to beta cells.
4. Formatting suggestions for easier readability:
 - a. Figure 1D- GeneHancer Enhancers to something like "all enhancers". And does this exclude the "Islet-Specific Enhancers?"
 - b. Figure 3B- Helpful to provide the denominator and % (e.g., 382/30665 [1.2%])
 - c. Figure 4A- Presumably there are more than one PDX1, NKX3-1, etc. sites than listed. What is the rationale for highlighting certain ones here?
 - d. Figure 4B- Helpful to provide the numerator and denominator and % in the figure
5. Clarification/error for top of page 7: "promoters, gene bodies and regions not mapping to promoters, genes or enhancers (red bars) were highly hypomethylated in insulinomas as compared to beta cells". Are the authors saying promoters not mapping to promoters? If that is the case, how do we know it's a promoter?
6. The difference between 42% and 47% is likely not significant. Suggest downplaying this point on page 8. "Remarkably, PDX1 and NFATC1 had the highest and best matching percentage of differentially methylated transcriptional regulator binding sites (14/30 sites or ~47% for both), compared to NKX3-1 (18/43 or ~42%)."
7. Since EndoC-betaH1 cells proliferate similar to insulinomas, discuss possible limitations to interpretation of Fig. 6.

Reviewer #3 (Remarks to the Author):

Karakose et al. report on methylome sequencing within the 11p15.5-15.4 region in 5 human beta cells and a set of 19 insulinomas. The study is mainly descriptive. An expected abnormal methylation pattern is observed at the insulin locus but no further experiment is done to confirm the findings and reveal the mechanisms behind such changes. This work represents a follow up of a paper published from the same authors in Nature Communication. The abnormal methylation pattern was already described, in a smaller dataset, in the previous work. The focus of the paper on the methylation of the 11p15.5-15.4 region limits the possibility to draw more general conclusions on global methylation changes that could be found in insulinomas.

The conclusion drawn in the study are highly speculative and not supported by the analysis performed. Overall the text is difficult to follow.

Specific comments:

The overall experimental design: "Three samples were pooled to achieve the amount needed for the bisulfite sequencing, and four others were sequenced individually."

The results obtained by pooling the 3 samples should be dropped from the analyses, pooling it with individual samples preclude the possibility of treating them as independent biological replicates.

Is not clear why subjects known to be members of MEN1 kindreds were intentionally excluded but, Ins_27 which is MEN1 mutated was finally included in the analysis.

In figure 1A the plot does not depict the inter-sample variability and CI of the methylation levels across samples. Are these differences statistically different?

Fig1B the clustering is not separating beta cells from insulinomas as claimed by the authors Ins18 is actually clustering with the beta cells. The most concerning aspect of this analysis is that the beta cells cluster is a subset of insulinomas and not as a separate group.

Failure of such analysis is critical to further conclusion made in the manuscript

Fig. 1C

Coloured regions – 'purple', 'dark red', 'light blue', 'pink'.

It's fine to use these colours in the plots but they are not very clearly defined. They are frequently referred to in the subsequent text without reference to their methylation status, requiring the reader to continually decipher what each section represents.

Fig2A the lack of correlation between promoter methylation and gene expression is surprising. The authors should rule out the possibility that this result is not driven by the heterogeneity of the sample replicates and tumor heterogeneity. The variation of gene expression across samples is not shown. Thus, further analysis are based on a highly speculative hypothesis that promoters of the genes involved in the locus may act as enhancers.

The analyses presented in Fig 4 5 and the overall model proposed is not supported by experiments allowing the 3D chromatin reconstruction in insulinoma or chip-seq experiments performed in insulinoma samples.

Response to Reviewers

Reviewer #1.

This study and a prior study by the same authors (Ref #10) have investigated the genomics and transcriptomics of human Insulinomas. The prior study found abnormal broad DNA hypomethylation in the imprinted chromosome 11p15.5-p15.4 region in insulinomas (n=10) compared to normal beta cells preps (n=2). The work was therefore extended to perform deep DNA methylome analysis of the same region in the current study using 9 additional insulinomas (total n=19), and 3 additional normal beta cells preps (total n=5).

Insulinoma-specific DNA methylation abnormalities in the 11p15 target region when analyzed together with transcription factor binding sites and enhancers predicted altered promoter and enhancer usage in insulinomas (in the 11p15 region). Also, published 3-D structural abnormalities in chromatin looping correlated with hypermethylated PDX1-binding sites at the INS gene in insulinomas indicating non-PDX1-mediated regulation of the INS gene.

Some of the conclusions/interpretations are speculative as mentioned at the end of each paragraph of the Discussion section for future studies, however, the results of this study will be useful to understand the basis of excessive proliferation

and insulin expression/secretion by the beta cells in insulinomas, with a potential to inform beta cell regeneration strategies and/or to modulate insulin gene expression.

Response: Thank you for the supportive comments and this nice summary, with which we agree. We appreciate them!

Some points to strengthen the interpretation and presentation of the work:

1a) Clarification about the level of insulin gene expression in human insulinomas: Discussion section page 11 bottom “INS gene expression is markedly ELEVATED in insulinomas compared to most normal cells”. Page 8 middle, “Collectively, these surprising events suggest that alternate transcriptional regulatory mechanisms must drive insulin gene OVEREXPRESSION in insulinomas”. And page 13 bottom, “Overall, these studies reveal a consistent, important and previously unrecognized pattern of 11p15.5-p15.4 methylation in human insulinomas. They suggest a model that may explain why INS is OVEREXPRESSED in insulin-producing PNETs”. However, NO CHANGE in INS expression is seen in Suppl Table-9 and current Fig. 2A (and text page 7 middle), and the same dataset in Ref #10 Fig 2b: Differentially expressed protein-coding genes in insulinomas (n=25) as compared to beta cells (n=22). “insulin mRNA, is EXPRESSED EQUALLY in beta cells and insulinomas.” Why are the interpretations different about insulin gene expression between the analysis in the current study and Ref #10? Are there other studies where elevated INS gene expression (at the RNA level) was observed in insulin-producing PNETs (insulinomas) compared to normal beta cells? Those References can be included in the manuscript.

Response: We agree that this is confusing and apologize. There are three key semantic issues here. First, both insulinomas and beta cells markedly overexpress, overproduce and over-secrete insulin at a level thousands-fold higher than any other normal cell type in the body, and any other benign or malignant human tumor cell type. Second, although insulinomas express, produce, and oversecrete insulin, the *INS* gene expression level in insulinomas is, on average somewhat lower than in normal beta cells (but still logs higher than any other cell type in the body). This is illustrated in the Table shown below in response to comment 4. Third, insulinomas comprise a small subset of a much larger group of pancreatic neuroendocrine tumors (PNETs). If a PNET produces insulin in sufficient quantities to cause hypoglycemia in the patient who harbors that PNET, then, by definition, it is an “insulinoma” subtype of PNET. If it is a PNET that produces clinically important quantities of gastrin (causing gastric or duodenal ulcers), somatostatin (producing diabetes), glucagon (also producing diabetes), then it is a “gastrinoma”, “somatostatinoma”, “glucagonoma”, respectively. If it produces no clinically relevant hormones it is a “non-functioning PNET”, which comprise the large majority of PNETs. Thus, by definition, insulinomas overexpress, over-produce and over-secrete insulin as compared to all other PNETs. We should have been clearer here and apologize. We have tried to clarify this confusing point by defining normal beta cells, non-functioning pancreatic neuroendocrine tumors (PNETs) and Insulinomas in the second paragraph of the Introduction. Thank you.

1b) Also, if there are any relevant References for the elevated level of NFATC1 gene expression in insulinomas they can be included in the manuscript.

Response: In mouse and human beta cells, there are four canonical NFATs, all of which are expressed at very low but physiologically important levels. They are important drivers and/or repressors of many beta cell transcription factors, phenotypic or specification markers, including *PDX1*, *GLUT2*, glucokinase, chromogranins A and B, islet amyloid polypeptide (IAPP) as well as *INS* (for a nice review, see Goodyer et al *Developmental Cell* 23:21, 2012). They also transcriptionally activate cyclin and cdk genes in beta cells and repress cell cycle inhibitors, in mouse and human beta cells. Unfortunately, since they are expressed at low levels in mouse and human beta cells, since they are multiple, and since ChIP-quality antisera to human NFATs are not available, their more general roles and genome-wide binding landscapes in human beta cells are poorly known.

We have re-checked the literature and found no papers describing alterations of the expression for any of the NFAT members in insulinomas. In our previous work (ref#10, Supplemental Table 8), we found changes in the expression for three of the four NFAT members, most notably NFATC1, also known as NFAT2, and had mentioned this in the original Discussion on page 12 in the paragraph on NFATs: “Thus, it was of great interest to observe that expression of *NFATc1*, as well as other NFAT family members, is increased in insulinomas¹⁰, and that its cognate response elements at the telomeric end of 11p15.5-p15.4 target sub-region are hypomethylated, and therefore presumably accessible to the NFAT family of transcription factors”. Together with the literature cited above on the role of NFATC1 in the regulation of the expression of *INS*, these observations support the involvement of NFATs in the determination of the insulinoma phenotype. Again, we have increased the emphasis on this point in the sentence above in the revision. Thank you.

chr	start	end	gene_symbol	mean expression (cpm)			logFC	pvalue	FDR
				beta cells	insulinomas	all samples			
chr18	77,155,856	77,289,325	NFATC1	0.5	4.1	1.5	3.093	8.14E-09	1.97E-07 **
chr20	50,003,494	50,179,370	NFATC2	21.7	17.1	19.0	-0.088	8.33E-01	8.87E-01
chr16	68,118,654	68,263,162	NFATC3	23.3	20.9	21.9	-0.617	1.11E-04	7.95E-04 **
chr14	24,834,879	24,848,810	NFATC4	10.4	14.4	12.5	1.114	7.47E-07	1.11E-05 **

2) Methods, RNA-Seq, relevant to the data presented in the Results section:

2a) How many of the 7 normal beta cells preps were subjected to RNA-Seq (pooled or individual preps)? Which sorting method was used (as per Ref #10) for the 7 beta cells preps: ZsGreen-sorted beta cells, Newport green-sorted beta cells or beta cells from insulin-antibody sorting?

Response: We agree that it is hard to follow the manner in which every sample was used, and its source. To help clarify this, we have developed a new Venn diagram figure (new **Supplementary Figure 1**) to make these issues clearer. In response to this specific question, none of the 7 normal beta cell preps used for methylome/bisulfite sequencing were submitted for RNAseq because the entire sample was used for bisulfite seq. The beta cell preps used for bisulfite seq were sorted by the Ad.RIP1-ZgGreen method described in reference 10. The beta cell preps used for RNAseq were prepared using the Ad.RIP1-ZsGreen method in our lab, by Newport Green sorting in the Nica-Dematzakis lab (new reference 32), and by insulin-immunolabeling and FACS sorting in the Bodgett lab (new reference 33). These RNAseq data sets are described in detail in reference 10. We have also expanded the first paragraph in Methods to make this point more clearly. Thank you.

2b) How many of the 19 insulinomas were subjected to RNA-Seq? “from 19 insulinomas, six had also previously been analyzed by RNA-Seq (Ins_11, Ins_13, Ins_19, Ins_22, Ins_23, Ins_24)”. However, Ins_19 is not listed from previous study, in Suppl Table 2. Does this mean that RNA-Seq data from only 5 insulinoma samples (not 6) was used in the current study? Were these 5 samples compared one to one for gene expression at the 11p15 locus with their respective 11p15 DNA methylation data?

Response: We apologize for an error here: the text should have read “(Ins_11, Ins_13, **Ins_18**, Ins_22, Ins_23, Ins_24)”. This list of insulinomas was correct in the original Supplementary Table 2, and has been corrected in the revised Methods/Results. This is also clarified in the new **Supplementary Figure 1** Venn diagram. Thank you for noticing this error.

In response to the question regarding gene expression, we also have added a new **Supplementary Figure 7** that describes the correlation between DNA methylation and gene expression for the six samples with both RNA seq and targeted bisulfite DNA seq data available. This figure confirms the cumulative data presented in the original **Figure 2A**. The differential methylation of promoters, irrespective of the different isoform of each gene is highly consistent across samples on average, with 5 of the 6 samples showing consistently either hypo- or hypermethylation. The differential methylation of promoters overall shows a reproducibility that is even higher than gene expression. These additional data are now provided in the Results section on page 7. Thank you.

2c) Gene expression data shown in Figure 2 and Figure 4. Not clear whether these Figures compare RNA-Seq data of all 25 insulinoma samples in Ref #10 with the DNA methylation data of 19 insulinoma samples in the current study? Note that among those 25 RNA-Seq insulinoma samples of Ref #10, only 5 have corresponding DNA methylation data. Also, regarding normal beta cells RNA-Seq, are those data from the 22 normal beta cells preps sequenced in Ref #10 or do they correspond to the 7 normal beta cells preps used in the current study?

Response: Again, we should have been clearer here. Throughout the paper, when exploring gene expression, we used the differential insulinoma-beta cell gene expression data from original reference 10, including 22 beta cell samples and 25 insulinomas, and included 5 new beta cell samples. In this new report, we describe bisulfite sequencing of 19 insulinomas, 9 of which were new in this study (see updated **Supplementary Table 2** for details). We have revised the Methods and Results, and have added a new **Supplementary Figure 1** to further clarify this point. Thank you for this comment.1

3) Comparing 3D chromatin structure data of embryonic pancreatic beta cell line (EndoC-βH1 beta cells) with adult beta cells (normal or insulinoma): Are there any differences expected in DNA methylation at the promoter or transcription factor

binding sites due to age-related methylation? Is there any known epigenetic difference between fetal/embryonic beta-cell INS gene regulation and adult beta-cell INS gene regulation?

Response: EndoC- β H1 cells are T-antigen- and Tert-immortalized human embryonic beta cells which are replete with large and multiple regions of chromosomal gain, loss and rearrangements (Lawlor, Stitzel et al., *Cell Reports* <https://doi.org/10.1016/j.celrep.2018.12.083>). There are no prior next-gene sequencing data on DNA methylation in human beta cells of any age. The only beta cell large, deep sequencing effort was reported by Avrahami et al (*Cell Metabolism* 22:619-32, 2015), but these were in whole mouse pancreatic islets (importantly, not pure sorted beta cells) comparing juvenile vs. older mice. Avrahami observed a relative and generalized age-related increase in methylation on a genome-wide basis, with a few focal regions of relative hypomethylation, notably in the regions of cell cycle inhibitors such as *cdkna2*, encoding p16ink4. Again, the data presented here are the first and only next-gen bisulfite sequencing data from either pure beta cells or insulinomas. This is an important part of the novelty and significance of this paper. We have highlighted this in final paragraph of the revised Introduction. Thank you.

4) Suppl Table 2: Insulin >50 units (in samples Ins_11, Ins_15, Ins_24, and Ins_32) and Insulin <10 units (in samples Ins_12, Ins_13, Ins_16, and Ins_22). Are there any extreme differences in INS or NFATC1 or PDX1 gene expression and 11p15.5-15.4 DNA methylation in these 8 samples with very high and low insulin?

Response: Thank you for this interesting question. We have checked the gene expression values for *INS*, *NFATC1* and *PDX1* for the samples with available RNAseq data (see Table below). Four samples, two with insulin < 10 units (Ins_11 and Ins_24) and two with insulin > 50 units (Ins_13 and Ins_22), were available for this analysis. No consistent differences were detected for *INS*, *NFATC1* and *PDX1* when comparing the two sample sets. From the DNA methylation point of view, Ins_11, Ins_15 (insulin < 10 units) and Ins_12, Ins_13 and Ins_16 (insulin > 50 units) all belong to Insulinomas Group 1 which is the group that shows the least variance in the distribution of the DNA methylation across samples. It is unlikely that these samples may show very different DNA methylation profiles at loci that may be associated with *INS*, *PDX1* or *NFACT1* expression. We speculate that this lack of correlation to the timing and nature of the measurements: the timing of the plasma insulin measurements in relation to meals, to treatment for hypoglycemia, etc. is not clear (values were obtained from chart review), and likely not standardized. In addition, insulin secretion by insulinomas may be episodic. We have not addressed this comment in the revision, because our response would be speculative and based on only very small sample numbers. If the Reviewer and Editor prefer, we could speculate on this issue in the Discussion. Thank you.

chr	start	end	gene_symbol	expression (cpm)								logFC	pvalue	FDR
				beta cells		insulin < 10u		insulin > 50u		insulinomas	all samples			
				AVG	ins_11	ins_24	ins_13	ins_22	AVG	AVG				
chr11	2,181,009	2,182,571	INS	30,696.4	17,754.1	16,420.7	824.8	15,076.2	8,203.4	14,660.8	-1.161	1.13E-01	2.09E-01	
chr13	28,494,157	28,500,368	PDX1	237.1	35.4	143.0	23.1	17.5	34.4	80.4	-1.157	3.65E-03	1.38E-02 **	
chr18	77,155,856	77,289,325	NFATC1	0.5	8.4	9.9	6.1	9.6	4.1	1.5	3.093	8.14E-09	1.97E-07 **	

5) Results, first paragraph bottom: Subject with germline MEN1 gene mutation: Is the mutation a VUS or obviously inactivating (stop, frameshift, etc.)?

Response: The SNV results in a Trp183Stop change. We have clarified this in **Supplementary Table 2**. Thank you.

Reviewer #2

This manuscript by Stewart and colleagues is an important follow up work to the Wang et al. 2017 paper. They have now increased the sample size of human insulinomas from ~10 to 38. Additionally they focused their efforts here on the methylation pattern of a critical imprinted region of chromosome 11 containing genes such as *INS*, *IGF2*. Interestingly they find that many of the islet specific enhancers are hypermethylated in insulinomas compared to beta cells. Their findings also suggest that insulinomas may possess a different mechanism than beta cells to express insulin. This reviewer finds this point particularly interesting and it should be expanded upon. There are some other conceptual and technical questions and comments that would strengthen this study and enhance its impact to the readers of Nature Communications.

Response: Thank you for these very supportive comments! We want to clarify that the original reference 10 included a total of 38 insulinomas, of which 26 had whole exome seq, 22 had RNAseq. 10 had targeted bisulfite sequencing of the 11p15 region. This is illustrated and clarified in a Venn Diagram in a new **Supplementary Figure 1**. The overall point for the current report is that we doubled the number of insulinomas (from 10 to 19) and beta cells (from 2 to 5) subjected to targeted next-

gen bisulfite sequencing, allowing meaningful statistical comparisons of beta cells to insulinomas for the first time. The necessary mélange of insulinomas and beta cells reflects the extreme rarity of insulinomas and the difficulty and expense of obtaining sufficient numbers human cadaveric islets and of FACS-purified human beta cells, as described in the comments to the Editor. We agree this is confusing and have tried to describe this more clearly in the Introduction, and in the new figure described above. Thank you.

Major comments:

1. It would be interesting to comment on the other differentially expressed genes in insulinomas outside of the 11p15 region now that they have expanded to 38 insulinomas. The previous study reported on ~25 insulinomas. In this study they also found 3 groups of insulinomas. Might this 11p15 region map or predict global gene expression? Directionality/causality would of course be hard to prove.

Response: This is a very important question about which we have thought a lot! As we describe, insulinomas display two major genomic/transcriptomic phenotypes: 1) recurring SNVs/MNVs/CNVs in genes that regulate chromatin organization and histone methylation/acetylation (e.g., Trithorax, Polycomb, SWI/SNF genes) throughout the genome; and, 2) abnormal methylation of the imprinted 11p15.5-p14.5 region. The question naturally therefore arises as to whether these two events are related. The short answer is that there are no data to address this important and interesting question.

Regarding the more specific question as to whether the 11p15.5-p14.5 imprinted region might influence gene expression in other, remote parts of the genome, Jian and Felsenfeld 2018, using in EndoC-βH1 human beta-cell-like cells, did indeed show that the area extending centromeric to the *INS* gene, the hypomethylated *SYT8/TNNI2* region, critically participates in *INS* gene expression, and in the chromatin organization of the whole genome of EndoC-βH1 cells (ref 25 of the original manuscript). Alterations of the methylation status of this area could plausibly have implications for genome-wide gene expression in insulinomas.

Conversely, for the question as to whether differentially expressed genes in insulinomas directly arise from the 11p15.5-p14.5 region, the answer would seem to be no, since there are some 3000 differentially expressed genes, most of which are derived from loci throughout the genome. Notably, most of the ~2000 genes that are upregulated in insulinomas are normally repressed in normal beta cells, in association with the repressive H3K27me3 histone mark (reference 10). With these thoughts in mind, we have addressed these interesting issues in the final paragraph of revised Discussion.

2. It possible, it would be compelling to see some data on global genomic methylation on the insulinomas compared to beta cells with this expanded set of insulinomas. Aside from the critical genes in 11p15, how unique or special is it compared to the rest of the genome?

Response: This is a really and interesting and important comment with which we completely agree.

In this vein, Libutti et al have reported on the genome-wide methylation effects *MEN1* mutations on a series of human parathyroid adenomas as compared to normal parathyroids and parathyroid adenomas not due to *MEN1* mutations. Intriguingly, the *MEN1* mutant human parathyroid glands display dramatic, genome-wide hypermethylation, events not observed in the several control groups. They go on to argue, using mouse genetic models, that *men1* inactivation leads to activation of *dnmt1* which in turn causes genome-wide methylation. It would be of great interests to perform comparable studies comparing human beta cells and insulinomas which are rich in *MEN1* and other Trithorax group mutations.

Unfortunately, insulinomas are very rare tumors and no one has performed genome-wide next-gen bisulfite sequencing, nor has anyone done this on purified human beta cells. We selected the 11p15 region for targeted bisulfite sequencing based on its well known imprinting status as detailed in the Introduction, and because of cost considerations. The studies in the current report generated approximately \$100,000 in sequencing costs and another \$100,000 in RNAseq, human islet purchase and FACS costs; we would love to expand to genome-wide studies but simply do not have the resources. We hope the Reviewer will agree that this would be an important goal for a future study if funding could be found: we would love to do this! This manuscript could well provide preliminary data for a grant that might support such studies. Finally, we would be happy to speculate and extrapolate on the relevance of the Libutti parathyroid adenoma studies described above to insulinomas of the Reviewer and Editor think this would be of value. Please advise. Thank you.

3. The authors state that methylation status surprisingly has weak correlation to gene expression in Figure 2A. However, there could be a threshold effect to the methylation status. The data shown could support a threshold phenomenon. Do the results change if a more absolute level of methylation is used rather than methylation fold change?

Response: We apologize for not having been clearer here. In **Figure 2A**, we focused on their directionality, i.e. hypermethylated promoter vs downregulated gene expression and hypomethylated promoter vs upregulated gene expression. It seems unlikely that, even by introducing a threshold, we would observe a change in directionality in our comparison. We welcome thoughts from the Reviewer on this point.

4. This study would greatly benefit from either finding out the molecular reason for the hypermethylation patterns in the region of interest. Additionally if any experiments could be done to reverse the methylation status, how this might change gene expression.

Response: We completely agree with this comment. Unfortunately, human beta cells only last a week or so in culture and do not replicate. Moreover, there are no truly representative human insulinoma cell lines with which to address this question using gene editing studies. There are EndoC- β H1 beta-like cells, but these cells are aneuploid and a poor model (Lawlor, Stitzel et al., *Cell Reports* <https://doi.org/10.1016/j.celrep.2018.12.083>). Human ES or iPSCs can be induced to differentiate along a beta cell lineage, but the final stages still represent immature human beta cells. Thus, it is hard to design and perform studies in human insulinomas to address this intriguing issue.

5. The authors state an interesting hypothesis that there could be non-PDX1 and NFATC1 dependent transcription of the INS gene for insulinomas. Any positive experimental evidence would be most welcome here. Alternatively, are there other examples in non-insulinoma conditions where NFATC1 could drive INS, such as beta cell failure, dedifferentiation, etc...

Response: Thank you for this comment with which we agree. Working with Drs. Yan Li, Xiaoxiao Liu and Fulai Jin at Case Western Reserve University, who are experts in Hi-C and other 3-D chromatin structural analyses, we have performed Hi-C studies to compare DNA looping profiles in FACS-sorted human beta cells, and insulinoma. We do indeed observe looping in the regions reported by Felsenfeld et al, as shown in the blue line in the original **Fig. 6A**, in both human beta cells and insulinomas, revealing contacts between the INS locus and the SYT8/TNNI2 locus, confirming the Felsenfeld data. Unfortunately, because of the need to use FACS-sorted human beta cells, the read depth of the Hi-C output is low and unable to clearly prove or disprove differences between insulinomas and beta cells predicted the model in **Fig. 6B**. Indeed, one most often uses large numbers of cells, in general cell lines, for the generation of these kinds of data, as did Felsenfeld et al. Conversely, no such data have been performed in human beta cells by ourselves or others. Thus, the creation of novel loops in the SYT8/TNNI2 region remains a model.

As an additional step to validating the model, we have also performed qPCR-ChIP studies to assess PDX1 binding to its normal PDX1 binding sites in beta cells and insulinomas. Indeed, as predicted by the model in **Fig. 6B**, PDX1 binds readily and reproducibly to its canonical binding sites in FACS-sorted human beta cells. Conversely, as also predicted, PDX1 fails to bind to hypermethylated PDX1 binding motifs in insulinoma, providing direct experimental support for this aspect of the model. This information is shown in a new **Supplementary Figure 8**, and discussed in the revised Result and Discussion. Thank you.

6. What is the protein expression of PDX1, NFAT, NKX3 in these insulinomas (especially for first 2)? Helpful to have some idea of protein expression and localization in insulinomas compared to beta cells.

Response: As we suggest in the original manuscript at the top paragraph on page 9, last line, the NKX3-1 findings are most likely incidental, so we have not pursued NKX3-1.

NFATs are far more interesting and likely important. As we had emphasized in the original manuscript, Goodyer and Kim (original ref 19, *Developmental Cell* 23:21-34, 2012) have shown that *Nfats* ChIP to the *Ins2* promoter in mouse beta cells, thereby inducing *Ins2* gene expression, as well as expression of canonical beta cell transcription factors and differentiation markers such as PDX1, IAPP, chromogranins A and B and glucokinase, summarized nicely in Figure 7E in that report. We have shown previously in reference 3 that NFATs are targets of the kinase DYRK1A, which phosphorylate NFATs and prevent their nuclear entry, and that drugs which inhibit DYRK1A signaling, enhance NFATs C1,2, 3, and 4 trafficking to the nucleus, and drive expression of cyclins and cdk. We have emphasized these studies in the revised Discussion on page 12, second to last paragraph.

We also agree that PDX1 is important. We have performed PDX1 ChIP studies at the *CDKN1C* locus. These confirm the ability of PDX1 to bind to key *CDKN1C* PDX1 regulatory sites in human islets, not in insulinoma, as shown a new **Supplementary Fig. 8**. We also have insulinoma samples and normal human pancreas samples available for immunohistochemistry to explore differences in PDX1 abundance in beta cells, as we have reported previously for p57 (reference 10, Fig. 5F). Unfortunately, because of the COVID-19 situation, we have not been allowed in our laboratories

since early March 2019, so have been unable to perform these studies. We will perform the PDX1 immunohistochemistry as soon as we are allowed to return to the laboratory. We believe that the manuscript is otherwise suitable for the second round of review, and PDX1 immunohistochemistry studies can be added on the next round of review when we are able to return to the lab. We hope that this will be an acceptable interim plan. Thank you.

Minor comments:

1. Low quality sequencing of insulinomas (e.g. #32)- if these are thrown out, do the insulinomas still cluster to the 3 groups?

Response: We had the same question ourselves while analyzing the original data. Briefly, there are two types of insulinomas that appear as “low quality” in **Figure 1B**. Samples 31, 32, 33 were of the lowest quality with respect to read depth. When these three samples are removed from the analysis, the grouping remains the same for insulinomas in groups 1 and 2 (see chart below). This is also apparent from the original **Figure 2C**, where the three groups are analyzed separately.

The other “apparent low quality” samples are Ins_23 and Ins_25. These two samples have chromosomal loss of approximately half of the imprinted 11p15.5-14.5 target sub-region resulting in lower total reads than the other samples. However, the read depth in the remaining 11p15.5-14.5 target sub-region was excellent and comparable to the others. We refer here to these samples as “low CpG call” samples. Removing these two insulinomas, together with Ins_32, only minimally altered the heatmap grouping. With a new Group 1 including samples from Ins_11 to Ins_17, and samples Ins_18 and Ins_19 now visually grouping within Group 2 with insulinomas Ins_21, Ins_22, Ins_24, Ins_26 and Ins_27. Considering the more peripheral association of Ins_18 and Ins_19 with other Group 1 samples in the full clustering analysis of **Figure 2B**, these results are not surprising and overall confirm the consistency of the visual grouping approach. A Group 3 can still be detected including samples Ins_31 and Ins_33 (see chart below) We have clarified this point in the revised Results at the top of page 6. Thank you.

2. Authors should comment on potential limitations with assessing beta cell heterogeneity within the insulinomas on interpreting their results.

Response: The Reviewer is correct in pointing out the heterogeneity among insulinomas: as described in reference 10, each insulinoma has a different mutation (SNV, MNV, Indel, CNV) profile, most of which occur in different chromatin modifying genes. As also described previously in reference 10, there is a surprising uniformity among insulinomas with regard to RNAseq, especially RNAs encoded by genes controlled by Trithorax and Polycomb genes. We see the same sort of striking uniformity here among CpG methylome profiles, despite distinct mutation profiles. Thus, “many roads lead to Rome” here: different mutational profiles lead to remarkably similar clinical phenotypes, as well as RNAseq and CpG methylation profiles. We are in the process of performing H3K4me3, H3K27me3, H3K27Ac ChIPseq and ATACseq on human beta cells and insulinomas, and comparing these with mutation landscapes as our next major effort. We have highlighted the commonalities among insulinoma clinical, pathological, RNAseq and methylome profiles, in the setting of a remarkably broad range of variants and affected genes in the Discussion, in paragraph 1 on page 11, and in the final paragraph on page 12. Thank you.

3. Figures 1D, 3C and throughout- make clear in the legend and figure that the comparison is to beta cells.

Response: Thank you for this helpful suggestion, which we have done.

4. Formatting suggestions for easier readability:

a. Figure 1D- GeneHancer Enhancers to something like “all enhancers”. And does this exclude the “Islet-Specific Enhancers?”

Response: We used the GeneHancer label advisedly. GeneHancer, although popular, represents only one of multiple available tools to score enhancers. Unfortunately, not all tools score enhancers the same way. Importantly, GeneHancer does not include Islet-Specific Enhancers as scored by Pasquali et al 2014 (ref 14 of the manuscript). Accordingly, in Figure 5B we showed that there is very limited overlapping between the two enhancer sets. For these reasons, we worry that changing the label of the bar graph in **Figure 1D** to “all enhancers” may suggest that we consolidated all calls of all “enhancer calling” tools. Thus, we feel that the “GeneHancer” is most appropriate label for displaying the results in Fig. 1D. We welcome the Reviewer’s thoughts on how best to present these data. Thank you.

b. Figure 3B- Helpful to provide the denominator and % (e.g., 382/30665 [1.2%])

Response: We also agree that this is complicated. The “n”s reported in the “All CpGs” panel refers to the cumulative number of transcriptional regulator within the 11p15.5-p15.4 target sub-region that carry at least 1 CpG. The “n’s” reported in the “Significant CpGs” panel indicates the cumulative number of transcriptional regulators that carry at least 1 statistically significant differentially methylated CpG. We have modified the figure by adding information on the number of both transcriptional regulators and their binding sites for both the “All CpGs” and the “Significant CpGs” charts. Thank you.

c. Figure 4A- Presumably there are more than one PDX1, NKX3-1, etc. sites than listed. What is the rationale for highlighting certain ones here?

Response: The goal of **Figure 4A** is to define the correlation between the expression and DNA methylation/rank enrichment, for those transcriptional regulators with binding sites mapping within the purple and dark red regions (see **Supplementary Figure 9** for the charts summarizing the results of the same analysis for light blue and pink regions). Thus, **Figure 4A** compares, for each transcriptional regulator identified in **Figure 3C**, the log-fold difference in expression (beta cell vs. insulinoma RNAseq from reference 10) with the average DNA methylation percent change across all binding sites within each color-coded region identified in **Figure 1C**. Thus, we are not selecting one specific site for each transcriptional regulator. Rather we are attempting to identify transcription factors and binding sites of particular importance in each of the four color-coded regions analyzed. We have tried to explain this more clearly in the revised Legend to Fig 4. Thank you.

d. Figure 4B- Helpful to provide the numerator and denominator and % in the figure 5.

Response: We have updated **Figure 4B** and **Figure 5** as requested. Thank you.

5. Clarification/error for top of page 7: “promoters, gene bodies and regions not mapping to promoters, genes or enhancers (red bars) were highly hypomethylated in insulinomas as compared to beta cells”. Are the authors saying promoters not mapping to promoters? If that is the case, how do we know it’s a promoter?

Response: Please see the response to comment 4a above. We realize that the wording of this sentence may be confusing. Here, we are considering six categories exactly as in **Figure 1D**: 1) islet-specific enhancers, 2) GeneHancer enhancers, 3) promoters; 4) gene bodies, 5) “regions that do not map to promoters, genes or enhancers” and 6) the full target region. We have modified the wording of this sentence at the junction of p. 6 and p.7 to make description clearer. Thank you.

6. The difference between 42% and 47% is likely not significant. Suggest downplaying this point on page 8. “Remarkably, PDX1 and NFATC1 had the highest and best matching percentage of differentially methylated transcriptional regulator binding sites (14/30 sites or ~47% for both), compared to NKX3-1 (18/43 or ~42%).”

Response: The Reviewer is correct that there is no single piece of evidence that supports exclusion of NKX3-1 from our model. However, our goal here was to summarize several different pieces of evidence that NKX3-1 had been incidentally included in our analysis. Among these, the percent of hypermethylated sites was just one of these points. The other supporting pieces of evidence are: 1) The *PDX1* and *NFATC1* differentially methylated transcriptional regulator binding sites were all hyper- or all hypomethylated, respectively, while *NKX3-1* showed a more mixed profile (4 hypo- and 14 hypermethylated sites); 2) No differentially methylated *PDX1* sites could be found that map to regions hosting *NFATC1* sites; and *vice versa*, *NKX3-1* sites mostly mapped within *PDX1*-rich regions and, in one case, within a *NFATC1*-rich region; and, 3) 12 out of 14 hypermethylated *NKX3-1* sites had different degrees of sequence overlapping with *PDX1* sites as curated by ReMap2018 (**Supplementary Table 16**). Accordingly, we concluded that: “Taken together, these observations most likely reflect incidental inclusion of *NKX3-1* in this analysis, and suggest *PDX1* and *NFATC1* as the key targets of the differential methylation of insulinomas”. Please also note that we have addressed this in the original Discussion on p. 12, second paragraph: “Among them, *NKX3-1* shares this region with *PDX1* and shows variable overlapping of its chromatin peaks as curated by ReMap2018. *NKX3-1* expression is also downregulated in insulinomas¹⁰. Whether the putative *NKX3-1* binding sites are a bioinformatic anomaly, or indicate sites employed by other transcription factors that overlap with the *PDX1* binding sites such as *NKX2-2* is unknown, and may suggest additional layers of complexity to the model in **Fig 6B**.” We believe this is a reasonable and balanced summary of the *NKX3-1* situation, but are happy to modify further if the Reviewer believes this would further enhance the Discussion. Thank you.

7. Since EndoC-betaH1 cells proliferate similar to insulinomas, discuss possible limitations to interpretation of Fig. 6.

Response: The bottom line for Endo-C BH1 cells is that although they are the best current human beta cell line available,

they are clearly not normal beta cells. First, they were immortalized using large T-antigen and Tert, which induce them to proliferate, albeit in clearly artificial ways: yes, they do proliferate, but their proliferation is in no sense “normal” or “physiological”. Second, they have multiple regions of chromosomal loss and gain, beautifully illustrated in karyotype data in Figure 1 in (Lawlor, Stitzel et al., *Cell Reports* <https://doi.org/10.1016/j.celrep.2018.12.083>). Thus, the proliferation in EndoC cells reflects large T-Ag overexpression, with resultant disruption of pRb enforcement of cell cycle arrest, likely enhanced by additional and multiple CNVs throughout the genome, and sheds little light on mechanisms that maintain arrest/quiescence in normal beta cells. This is precisely why we have taken so much trouble to compare insulinomas to normal, quiescent beta cells. We have added a discussion of the limitation of extrapolating from Endo-C cells to normal beta cells in the final paragraph of the Discussion. Thank you.

Reviewer #3

Karakose et al. report on methylome sequencing within the 11p15.5-15.4 region in 5 human beta cells and a set of 19 insulinomas. The study is mainly descriptive. An expected abnormal methylation pattern is observed at the insulin locus but no further experiment is done to confirm the findings and reveal the mechanisms behind such changes. This work represents a follow up of a paper published from the same authors in Nature Communication. The abnormal methylation pattern was already described, in a smaller dataset, in the previous work. The focus of the paper on the methylation of the 11p15.5-15.4 region limits the possibility to draw more general conclusions on global methylation changes that could be found in insulinomas. The conclusion drawn in the study are highly speculative and not supported by the analysis performed.

Response: We completely agree with this summary, and the importance of generating direct experimental confirmation of the bioinformatic predictions. Reviewer 2 in comment 2 made a similar point. It is important to understand that pancreatic islets comprise ~1% of total pancreas mass, and among islets, beta cells account for ~40% of total cells. Thus, beta cells comprise some 0.4% of pancreatic cells. As the Reviewer will understand, and as described in more detail in response to comment 2 below, obtaining sufficient numbers of beta cells for studies such as these is difficult, and losses of beta cells during islet isolation from the whole pancreas, and subsequent beta cell purification by FACS, are enormous. This is why the largest NGS “islet” datasets are obtained using the Endo-C β H1 cell line, a large T-antigen- and Tert-immortalized, and very aneuploid human beta cell-like line (Lawlor, Stitzel et al., *Cell Reports* <https://doi.org/10.1016/j.celrep.2018.12.083>). Please also see the response to Reviewer 2, comment 7, above, and comment 3 below. For these reasons, there are no data on genome-wide methylation in authentic human beta cells, nor are there likely to be any such data in the near future: it simply is not possible to obtain sufficient numbers of pure human beta cells for genome-wide methylome analysis. Seen from this perspective, we hope the Reviewer will agree that what we have accomplished is something of a minor miracle, and we have taken the work as far as is possible with current technology. We have added a comment to this point in last paragraph of the revised Discussion. Thank you.

Overall the text is difficult to follow.

Response: We apologize for this. It is a complicated study, and difficult to describe. We have tried to enhance clarity throughout in response to each of the three Reviewers’ comments. Additional specific suggestions to improve clarity are welcome!

Specific comments:

1) The overall experimental design: “Three samples were pooled to achieve the amount needed for the bisulfite sequencing, and four others were sequenced individually.”

Response: We have tried to clarify this language in the first paragraph of Methods and a new **Supplementary Fig. 1**. Thank you.

2) The results obtained by pooling the 3 samples should be dropped from the analyses, pooling it with individual samples preclude the possibility of treating them as independent biological replicates.

Response: Again, obtaining sufficient numbers of human cadaveric islets, and FACS-sorting sufficient number of human beta cells is challenging and expensive, as described in Hart and Powers (*Diabetologia* 62:212-22, 2019). Please note that the collection of RNAseq data from 25 sets of human beta cells in original reference 10 is the largest human beta cell RNAseq dataset ever reported, by far. Please also understand that while the amount of RNA required for RNAseq is trivial

and relatively easy to obtain from FACS-sorted beta cells, in contrast, the amount of DNA required for targeted bisulfite sequencing of the relatively tiny 11p15 region is substantial (~500 ng), near the maximum obtainable from FACS-sorted beta cells. As a result, for the current study, all of the FACS-sorted beta cells collected were used to prepare DNA for methylome/bisulfite sequencing, and in some cases was insufficient. This resulted in a need for pooling three samples to obtain enough DNA for one set of beta cells, and in all cases, no RNA was prepared for RNA seq from the bisulfite-seq samples. We are comfortable with this approach, since the bisulfite sequencing profiles in Fig. 1B of the pooled sample (Beta_2) grouped together with the other four beta cell samples. Similarly, the component analysis of **Supplementary Figure 3** (panels C and D) shows that sample Beta_2 behaves like the other 4 beta cell samples.

Further, we have rerun the statistical analysis by excluding sample Beta_2 and the results are strikingly similar to those obtained by using all five beta cell samples (see Figure below). The number of statistically significant differentially methylated CpG dinucleotides in both analyses is highly similar. The exclusion of Beta_2 leads to the loss of only 320 statistically significant differentially methylated CpG dinucleotides, which is remarkable considering the reduction of 25% in the sample size of the beta cell group. Additionally, some 1,950 statistically significant differentially methylated CpG dinucleotides still overlap between the two analyses further supporting the similarity between Beta_2 and the other four beta cell samples.

Finally, we also note that Reviewers 1 and 2, and all four of the Reviewers of Reference 10 were comfortable with this approach. We welcome suggestions from the Reviewer and Editor for alternate approaches with this background in mind. Thank you.

3) Is not clear why subjects known to be members of MEN1 kindreds were intentionally excluded but, Ins_27 which is MEN1 mutated was finally included in the analysis.

Response: We apologize for not having been clearer here. In original study in reference 10, we excluded people with insulinomas who were known to be members of a MEN1 kindred, since it is well known that MEN1 SNVs/MNVs/Indels are causally associated with insulinomas and other PNETs (original references 9,30,31), and we did not wish to re-learn that lesson: we wanted to identify novel SNVs/MNVs/CNVs associated with insulinoma pathogenesis. Despite our efforts to exclude MEN1 subjects in reference 10, however, one subject did prove to have a MEN1 protein altering mutation, which also proved to be germline. A second subject was discovered to have a somatic protein-altering *MEN1* mutation. By chance, neither of these had DNA bisulfite sequencing.

In the current study, there is one subject (Ins_27) with a *MEN1* SNV that results in a Trp183Stop change. We have clarified this issue in the Methods (first paragraph), Results (first paragraph) and **Supplementary Table 2**. Thank you.

4) In figure 1A the plot does not depict the inter-sample variability and CI of the methylation levels across samples. Are these differences statistically different?

Response: This a great question. We now provide a revised version of this Figure, and its legend, showing shaded versions of the two methylation tracks illustrating 95% confidence intervals. We believe this new format makes the broad regional hypomethylation even more striking and apparent. Thank you for this suggestion.

5) Fig1B the clustering is not separating beta cells from insulinomas as claimed by the authors Ins18 is actually clustering with the beta cells. The most concerning aspect of this analysis is that the beta cells cluster is a subset of insulinomas and not as a separate group. Failure of such analysis is critical to further conclusion made in the manuscript.

Response: We agree with this comment, and had clearly indicated, in the original Results paragraph 4, that the clustering is more visual than statistically significant. We used this approach because we visually detected differences in our samples that were not identified when conducting the traditional clustering analysis. For the clustering analysis, we used well-validated dimensional and component analysis methods to estimate the number of clusters for our samples (see **Supplementary Figure 3**). These methods determined that the “best” cluster number for our samples was actually “one” (see **Supplementary Figures 3A, C and D**), suggesting that statistically meaningful clusters did not exist in our data set. Moreover, the statistical “by group” analysis in **Figure 1C** and **Supplementary Figure 4** further demonstrate the lack of meaningful clusters among insulinomas. At the same time, the data in **Figures 1A** and **1C** make it clear that beta cells and insulinomas are distinct with respect to methylation patterns, and insulinomas are more similar to each other in component analysis than they are to beta cells (**Supplementary Figures 3C and D**). Finally, the heatmap of **Figure 1B** clearly demonstrates that beta cells are the most cohesive sample set, while insulinomas show higher degree of variability in their DNA methylation profiles. This heatmap, as it refers to distance across sample sets, cannot be used, alone, to determine functional relationships between sample sets. With these considerations in mind, we have not changed the text, but welcome suggestions from the Reviewer and Editor.

6) Fig. 1C. Coloured regions – 'purple', 'dark red', 'light blue', 'pink'. It's fine to use these colours in the plots but they are not very clearly defined. They are frequently referred to in the subsequent text without reference to their methylation status, requiring the reader to continually decipher what each section represents.

Response: We agree. We now provide the precise coordinates of the four colored regions in the text where they are first introduced in Results, and in the legend to **Figure 1**. We also precede them when used later in the text by “hypermethylated” or “hypomethylated”, as appropriate. Thank you.

7) Fig 2A the lack of correlation between promoter methylation and gene expression is surprising. The authors should rule out the possibility that this result is not driven by the heterogeneity of the sample replicates and tumor heterogeneity. The variation of gene expression across samples is not shown. Thus, further analysis are based on a highly speculative hypothesis that promoters of the genes involved in the locus may act as enhancers.

Response: We also were surprised by the lack of correlation between promoter methylation and gene expression described in **Figure 2A**. Please also see also the response to comment 4 above which makes it clear that that the abnormal methylation patterns in insulinomas were not random, but very consistent among all of the insulinomas. This, together with the striking but very reproducible difference in methylation in the 11p15.5-p14.5 region between beta cells and insulinomas, and knowing that this region was imprinted and functions abnormally in Beckwith-Weideman syndrome with beta cell hyperplasia and hypoglycemia, were the fundamental observations that led us to consider that DNA looping in this region might be abnormal in insulinomas compared to beta cells, and to take a deeper look at TF binding sites and TF gene expression. This is exemplified by the analysis of data generated by the Roadmap Epigenomics data in **Figure 2B**, which clearly shows that the promoter regions of 24 of the 28 genes that we analyzed, carry either enhancer or dyadic histone signatures across multiple cell lines and tissues. Taken together with our findings that predict an important role for islet-specific enhancers in insulinomas (see **Figures 1D, 5A and 5C**), this concept is further supported by the chromatin loops described Jian and Felsenfeld in EndoC-βH1 cells (ref 25 of the original manuscript) illustrating that the area extending centromeric to the *INS* gene, including the *SYT8/TNNI2* region contacts the *INS* promoter/enhancer region and participates in *INS* gene regulation. We have not addressed this comment in the revised manuscript because we hope it is clarified in earlier comments, but would be happy to address this further as recommended by the Reviewer. Thank you.

8) The analyses presented in Fig 4 5 and the overall model proposed is not supported by experiments allowing the 3D chromatin reconstruction in insulinoma or chip-seq experiments performed in insulinoma samples.

Response: Again, we agree completely. Please see our comments to the Editor above, and response to Reviewer 2, comment 5. We have tried hard to develop 3-D chromatin studies as discussed above, but these also require larger numbers of FACS-sorted human beta cells. In a new **Supplementary Figure 8**, we have now added PDX1 qPCR data supporting the concept that PDX1 binding to its canonical binding sites upstream of the INS gene are present in FACS-sorted beta cells, but absent insulinoma DNA in the same region. This provides some direct experimental support for the model predicted by the bioinformatic analysis. Thank you.

We believe the manuscript has been substantially strengthened by the Reviewers' helpful and supportive comments. We understand that the study and its narration are necessarily complex, but the advances are highly novel and important in the world of diabetes and beta cell biology. We appreciate the Reviewers' work in reviewing and improving the manuscript.

REVIEWER COMMENTS

Reviewer #1 (Remarks to the Author):

I have no additional comments for the revised version.

Reviewer #2 (Remarks to the Author):

Overall the manuscript is much clearer and easier to read. Supplementary figure 1 is helpful. Most questions have been satisfactorily addressed with just a couple remaining.

Responses to Major Comments:

1. No further questions.
2. No need to speculate with parathyroid adenomas with MEN1 mutations since they are a different cell type.
3. The data presented shows methylation log fold change in insulinomas compared to beta cells. For example, let's say a region in an insulinoma is heavily methylated. The fold change of methylation compared will be quite different if a region starts out as heavily or lightly methylated in beta cells. Since different regions will have different starting methylation statuses, the methylation fold change may not adequately capture the starting point. Yes this gets complex as the same argument could be applied to levels of gene expression.
4. No further questions.
5. No further questions.
6. Agree that the PDX1 IHC would be helpful before final publication.

Responses to Minor Comments:

1. No further questions.
2. No further questions.
3. No further questions.
4. No further questions.
5. No further questions.
6. Is the point that NKX3-1 is used as a reference control?
7. No further questions.

Reviewer #3 (Remarks to the Author):

In the updated version of the manuscript the Authors addressed many of the issues raised in the revision. Yet, in the reviewer opinion, one points still need to be addressed precluding the possibility of this work to be published in Nature Communication.

Referring to the clustering in Fig 1B the authors state that "the clustering is more visual than statistically significant. We used this approach because we visually detected differences in our

samples that were not identified when conducting the traditional clustering analysis.”

This result is then used to generate further analyses and lead to several conclusions. Thus, robust statistical data supporting these analysis should be provided.

Reviewer #4 (Remarks to the Author):

Karakose et al. compared DNA methylation status of human insulinomas and pancreatic islet beta cells and found abnormal methylation patterns within 1.35 megabase region at 11p15.5 -15.4. Combined with previously published 4C-seq data and ATAC-seq data, they indicated the enhancer switch for insulin gene and suggested NFAT as a potential transcription factor in insulinoma. However, their conclusions are not fully validated and mostly speculative, so may require additional experimental and informatic analysis.

1. Prediction of the transcriptional factor from the binding motifs may require careful investigation. Canonical beta cell transcription factors such as PDX1, NKX6.1 are highly expressed in pancreas, while NKX3.1 expression is specific to prostate and testis. The authors stated that NKX3-1 is downregulated in insulinoma in the reference #10, which I could not confirm. You need to show NKX3.1 data from the Ref. 10 in the supplementary Table 15. Furthermore, NKX2-2 is known to be involved in the development of insulin-producing beta cells in the endocrine pancreas. In Ref #14, where NKX3.1 and NKX2.2 ChIP-seq data were also obtained together with PDX1 (ERP004003), NKX2.2 binding sites are well overlapped with those of PDX1. However, NKX2.2 binding motif (TTAAGTACTT) is similar to that of NKX3-1 (TAAGTA). If you would like to propose the role of NKX3.1 binding sites in the 'red' region in your manuscript, you may need to show a reason to do so.
2. The same is the case with NFAT. The Role of NFATc Signaling in Postnatal β Cell Development is well studied. First, they need to determine which NFATc protein, namely NFATc1 or 2, is recruited to the telomeric enhancer. Keller MP, et al. reported that NFATc2 is important in beta cell proliferation (<https://doi.org/10.1371/journal.pgen.1006466>), while NFATc1 is implicated in the text and Figures 4 and 6. Furthermore, how NFATc protein remains to be localized or mislocated at the telomeric enhancer in insulinoma would be of great interest. Besides PDX1 (Supplemental Fig. 8), they should perform NFATc ChIP assay to validate their proposed model. Of course, sample preparation could be an issue. Several sensitive chromatin assays, such as CUT&RUN, could be useful.
3. 11p15 is well known as the imprinted region and imprinting abnormalities are associated with beta cell proliferation in the focal variant of congenital hyperinsulinism and Beckwith–Wiedemann Syndrome. Loss of heterozygosity (genomic) and loss of imprinting (epigenomic) have been reported in this region. Since they performed bisulfite sequencing over 1.35 megabase region, they should be able to make very fine assessment on genomic and epigenomic alteration for each case, which cannot be clear from Figure 1. I suppose we should see 50% methylation level in the imprinted region for normal beta cell samples.
4. To propose a model in Figure 6, CTCF binding need to be examined to show the change of 3D looping. Most of the CTCF binding sites are shared among various cell types, but it would be ideal to examine the CTCF binding sites in EndoC- β H1 cells.
5. Given that there is heterogeneity in methylation status, you may need to compare the gene expression individually, not as a group.

Minor points:

- In the heatmap of DNA methylation data in Figure 1B, they may get better clustering pattern by selecting a set of variable CpG sites.
- In Figure 2, they divided the genome into 100-kb bins and showed the percentage of chromatin status, which is too coarse to discuss promoter or enhancer position for gene-rich 11p15 regions.

REVIEWER COMMENTS

Reviewer #1:

"I have no additional comments for the revised version."

Response: Thank you for your careful and thoughtful review on the first round of review, and for your help with the second round. We greatly appreciate it!

Reviewer #2:

"Overall the manuscript is much clearer and easier to read. Supplementary Figure 1 is helpful. Most questions have been satisfactorily addressed with just a couple remaining."

Response: We thank the Reviewer for their hard work in the review process and very helpful feedback. See specific responses below.

"Major Comments:"

1. *"No further questions."*

Response: Thank you.

2. *"No need to speculate with parathyroid adenomas with MEN1 mutations since they are a different cell type."*

Response: Thank you for this recommendation. We have therefore not discussed the parathyroid, MEN1 hypermethylation story in the revised manuscript.

3. *"The data presented shows methylation log fold change in insulinomas compared to beta cells. For example, let's say a region in an insulinoma is heavily methylated. The fold change of methylation compared will be quite different if a region starts out as heavily or lightly methylated in beta cells. Since different regions will have different starting methylation statuses, the methylation fold change may not adequately capture the starting point. Yes this gets complex as the same argument could be applied to levels of gene expression."*

Response: Thank you for this comment which we believe applies to Figure 2A. We agree that the methylation fold-change and expression fold-change do not provide a precise depiction of their respective baseline statuses. On the other hand, we do believe this is a standard way to present and analyze these types of data. Variations in promoter methylation spanning the whole 0 to 100% range have been shown for both high and low expressed genes (Bell JT, Pai AA, Pickrell JK, Gaffney DJ, Pique-Regi R, Degner JF, Gilad Y, Pritchard JK. DNA methylation patterns associate with genetic and gene expression variation in HapMap cell lines. *Genome Biol* 12(1):R10; 2011), although the reasons for this broad range are still incompletely understood (reviewed in Greenberg MVC, Bourc'his D. The diverse roles of DNA methylation in mammalian development and disease. *Nat Rev Mol Cell Biol*; 20(10):590-607; 2019). For this reason, Jiao et al and Wreczycka et al recommend using differential DNA methylation to study its correlation with changes in gene expression in the manner we have employed (Jiao Y, Widschwendter M, Teschendorff AE. A systems-level integrative framework for genome-wide DNA methylation and gene expression data identifies differential gene expression modules under epigenetic control. *Bioinformatics*; 30(16):2360-6; 2014. Wreczycka K, Gosdschan A, Yusuf D, Gruning B, Assenov Y, Akalin A. Strategies for analyzing bisulfite sequencing data. *J Biotechnol*; 261:105-15; 2017). Conversely, we are not aware of these types of data being presented as X- and Y-axis linear data, and doing so would lose the reference standard (blue line in Fig 2A) for display purposes. Figure 2B and the

Supplemental Tables also display and confirm differential methylation and expression data for individual loci in Fig. 2A for interested readers, so we believe they are accurate and meaningful. We certainly welcome specific suggestions as to alternate methods to analyze and display these data. Thank you.

4. “No further questions.” Response: Thank you.

5. “No further questions.” Response: Thank you.

6. “Agree that the PDX1 IHC would be helpful before final publication.”

Response: This question arises because we report in the Results on the junctions of pages 8-9 and 9-10 and Figures 4,6 that PDX1 mRNA expression in insulinomas, while abundant, is lower than in normal FACS-sorted beta cells (-1.157 log-fold change). This is of interest because we show that its cognate binding sites upstream of the *INS* locus are hypermethylated in Figs 4-8, and inaccessible to PDX1. So it was of interest to see whether PDX1 protein abundance is reduced in insulinomas.

With this goal in mind, during the COVID lab shutdown, we were able to obtain human pancreas sections from three patients undergoing insulinoma resection, and whose insulinoma surgical specimen also contain normal adjacent pancreas. We did this so that the adjacent normal islets with their normal beta cells could serve as a normal control for PDX1 intensity in the same immunohistochemistry experiment, same tissue section, same fixation, same antibodies and titers, etc. We have immunolabeled them for PDX1 and insulin, as shown in the Figure below, and can clearly see PDX1 in beta cells and insulinomas, but cannot discern quantitative differences in PDX1 abundance by immunohistochemistry. We interpret this to mean that although PDX1 mRNA expression is lower on average in insulinomas than in normal beta cells, this quantitative difference is too small to be apparent by the quantitatively crude immunohistochemistry measure, or that PDX1 mRNA and PDX1 protein quantities may differ as a result of PDX1 translation efficiency or stability, or that the three insulinoma/normal islet pairs we elected to immunolabel are not representative of the 22 sets of beta cells and 19 insulinomas on which we performed RNAseq.

None of these interpretations alter anything in the manuscript. The big picture here relates not to PDX1 protein abundance, but rather to the inability of endogenous PDX1 in insulinomas to bind to its cognate binding sites upstream of the *INS* gene because they are hypermethylated. This is a key point, and one that we have confirmed experimentally by qPCR in Supplementary Figure 8. Since the PDX1 immunohistochemistry results do not affect any of the results or interpretations in the study, we have elected to leave them out of the manuscript, but would be very happy to include the figure above and relevant text as a Supplementary Figure if the Reviewer or Editor so desire. Please advise. Thank you.

“Minor Comments:”

1. “No further questions.” Response: Thank you.

2. “No further questions.” Response: Thank you.

3. “No further questions.” Response: Thank you.

4. “No further questions.” Response: Thank you.

5. “No further questions.” Response: Thank you.

6. “Is the point that NKX3-1 is used as a reference control?”

Response: We apologize for not being clearer. New Reviewer 4 had a similar question (see below). Our interpretation of the PDX1/NKX3-1 binding locus findings is that NKX3-1 is an innocent bystander captured in the enrichment analysis. That is, we believe that the “hypermethylation signal” for NKX3-1 binding loci is coming from PDX1 loci. We say this because the multiple hypermethylated PDX1 and NKX3-1 binding loci identified in Fig. 2C (now 4C) largely overlap with one another, as highlighted in Fig. 6B and 7B. In further support of this interpretation, NKX3-1 is not recognized as being an important transcription factor in beta cells (in contrast, for example, to PDX1, NKX6-1 and NKX2-2, which are essential beta cell transcription factors). We have tried to clarify this in the text in several places, as summarized in response to comment 1 from new Reviewer 4 below. Thank you.

7. “No further questions.” Response: Thank you.

Reviewer #3:

“In the updated version of the manuscript the Authors addressed many of the issues raised in the revision. Yet, in the reviewer opinion, one points still need to be addressed precluding the possibility of this work to be published in Nature Communication.”

Response: Once again, we want to thank the Reviewer for their effort and helpful feedback in the Review process. We appreciate the hard work and helpful feedback.

1. *“Referring to the clustering in Fig 1B the authors state that “the clustering is more visual than statistically significant. We used this approach because we visually detected differences in our samples that were not identified when conducting the traditional clustering analysis.” This result is then used to generate further analyses and lead to several conclusions. Thus, robust statistical data supporting these analysis should be provided.”*

Response: This is confusing, and Reviewer 4 had a similar question (see below). Again, we apologize for not being clearer. We did indeed say the clustering was “*more visual than statistically significant*”, but the results were definitely not “*then used to generate further analyses and lead to several conclusions.*” We compared insulinomas to normal beta cells based on their clearly distinct biology and pathobiology. Indeed, we went to some lengths to show that these data were not appropriate for use in further analyses (Supplementary Figures 3A-D, and the “Groups 1, 2 and 3” tracks in Fig. 1C), and we did not use the clustering data for analyses in the remainder of the manuscript. See also minor comment 1 by Rev 4 below. We have tried to clarify the confusing language in the Revision on the top paragraph on page 6 where we write: “Going forward, the two insulinoma and beta cell datasets were compared based on their representing normal and abnormal biology.” Thank you.

Reviewer #4 (New Reviewer):

“Karakose et al. compared DNA methylation status of human insulinomas and pancreatic islet beta cells and found abnormal methylation patterns within 1.35 megabase region at 11p15.5 -15.4. Combined with previously published 4C-seq data and ATAC-seq data, they indicated the enhancer switch for insulin gene and suggested NFAT as a potential transcription factor in insulinoma. However, their conclusions are not fully validated and mostly speculative, so may require additional experimental and informatic analysis.”

Response: Thank you for your careful review and helpful comments. We appreciate these and the effort they entailed!

Note that we have broken Comment 1 into two parts below for clarity of response.

1a. *“Prediction of the transcriptional factor from the binding motifs may require careful investigation. Canonical*

beta cell transcription factors such as PDX1, NKX6.1 are highly expressed in pancreas, while NKX3.1 expression is specific to prostate and testis”.....”Furthermore, NKX2-2 is known to be involved in the development of insulin-producing beta cells in the endocrine pancreas. In Ref #14, where NKX3.1 and NKX2.2 ChIP-seq data were also obtained together with PDX1 (ERP004003), NKX2.2 binding sites are well overlapped with those of PDX1. However, NKX2.2 binding motif (TTAAGTACTT) is similar to that of NKX3-1 (TAAGTA).” ... “If you would like to propose the role of NKX3.1 binding sites in the ‘red’ region in your manuscript, you may need to show a reason to do so.”

Response: We completely agree that there is no reason to propose an important role for NKX3-1 in beta cell or insulinoma biology and did not intend to suggest that. We had tried to make this point in the Results on page 9, second paragraph: “Taken together, these observations most likely reflect incidental inclusion of NKX3-1 in this analysis, and suggest that PDX1 and NFATC1 binding sites as the key targets of the differential methylation of insulinomas,” and the top of page 10: “All but one site were shared by the two transcriptional regulators (meaning PDX1 and NKX3-1), again suggesting: 1) incidental inclusion of NKX3-1.” Please also see the response to Reviewer 2, minor comment 6 above, where a similar question was raised. We definitely do not believe that NKX3-1 binding loci play a physiological or pathophysiological role in beta cell or insulinoma biology. We have tried to say this more clearly in the revised Results on page 9, top and bottom paragraphs, and Discussion, page 12, paragraph 3. Thank you.

1b. The authors stated that NKX3-1 is downregulated in insulinoma in the reference #10, which I could not confirm. You need to show NKX3.1 data from the Ref. 10 in the original supplementary Table 15.

Response: The NKX3-1 expression data are present in Suppl. Table 8 in ref 10. We now have added them to the PDX1 and NFATC1 differential expression data in Suppl. Table 16 in the revised manuscript, which is shown below:

Supplementary Table 16. Expression values for NKX3-1, PDX1 and NFATC1 in beta cells and insulinomas from Wang et al, 2017 (see ref #10)

chr	start	end	gene_symbol	mean expression (cpm)			logFC	pvalue	FDR
				beta cells	insulinomas	all samples			
chr8	23,536,206	23,540,450	NKX3-1	3.8	0.9	1.7	-2.301	1.86E-06	2.44E-05 **
chr13	2,181,009	2,182,571	PDX1	237.1	35.4	80.4	-1.157	3.65E-03	1.38E-02 **
chr18	77,155,856	77,289,325	NFATC1	0.5	4.1	1.5	3.093	8.14E-09	1.97E-07 **

2. “The same is the case with NFAT. The Role of NFATc Signaling in Postnatal β Cell Development is well studied. First, they need to determine which NFATc protein, namely NFATc1 or 2, is recruited to the telomeric enhancer. Keller MP, et al. reported that NFATc2 is important in beta cell proliferation (<https://doi.org/10.1371/journal.pgen.1006466>), while NFATC1 is implicated in the text and Figures 4 and 6. Furthermore, how NFATc protein remains to be localized or mislocated at the telomeric enhancer in insulinoma would be of great interest. Besides PDX1 (Supplemental Fig. 8), they should perform NFATc ChIP assay to validate their proposed model. Of course, sample preparation could be an issue. Several sensitive chromatin assays, such as CUT&RUN, could be useful.”

Response: Here, we also agree. The NFAT story is interesting and important, but it is also very complicated. We have published several papers (references 3 and 4, as well as *Science Translational Medicine* PMID: 32051230) and cited others (reference 2) demonstrating that NFATs play an important role in controlling human beta cell proliferation and differentiation. We also agree that it would be important to demonstrate that NFATs ChIP to the SYT8/TNNI2 region in insulinomas, but not beta cells. Coincidentally, we know the Keller/Attie group well and have published papers with them, and they are well aware of our work, as Dr. Attie visited our lab last October, and we have discussed the issues with Dr. Keller by phone. As an example, the Keller paper referred to by the Reviewer cites our work on NFATs as their reference 36.

Here are the problems we confront. First, there are four NFATs, all of which are expressed at low levels in human beta cells and all of which share identical or similar DNA binding motifs. Second, there are no ChIP-quality

antisera for the four NFaTs. And third, we must work with human beta cells (as the essential normal control) which must be FACS-sorted from human cadaveric islets. We have been able to perform histone mark ChIPseq (H3K4me3, H3K27me3, H3K27Ac) and ATACseq on small numbers (~6000) of FACS-purified human beta cells, but performing transcription factor (TF) ChIPseq is more difficult, and neither we nor others have been able to do this for any NFaT in human beta cells. We agree that cut-and-run holds promise and are trying to do that at present with several TFs of interest including NFaTs.

The Keller/Attie group confronted these same issues in the reference above. They approached them by adenoviral overexpression in mouse islets of heavily mutated constitutively active mouse NFaTs (12 and 17 serial mutations in mouse NFaTc2 and mouse NFaTc1, respectively). Moreover, they used the potent but non-selective CMV promoter, instead of a cell-type specific promoter (insulin promoter), preventing clear interpretation as to which islet cell types events are occurring in, as they acknowledge in their Discussion section. While we believe that this is beautiful work, one must also agree that the system is artificial in several senses, and may or may not reflect authentic events in human beta cells.

For us, the conundrum is whether we should do ChIP or ChIPseq using non-native, artificially overexpressed, epitope-tagged and/or constitutively active NFaTs, or wait until better antisera and smaller cell number ChIP/ChIPseq/cut-and-run methods are available. We have chosen the second path, which we hope the Reviewer will agree is most reasonable. We are currently focusing our technical efforts on what we believe is a more pressing problem: defining the chromatin looping in this region using small cell number, 3-D chromatin-capture methods. We hope the Reviewer and Editor will agree that this is the most reasonable choice.

With the Reviewers' comments in mind, we have further emphasized the hypothetical nature of the NFaT involvement in the revised Discussion, and have added the Keller/Attie reference (new reference 31) to the text on page 12, paragraph 4. Thank you.

3. *"11p15 is well known as the imprinted region and imprinting abnormalities are associated with beta cell proliferation in the focal variant of congenital hyperinsulinism and Beckwith–Wiedemann Syndrome. Loss of heterozygosity (genomic) and loss of imprinting (epigenomic) have been reported in this region. Since they performed bisulfite sequencing over 1.35 megabase region, they should be able to make very fine assessment on genomic and epigenomic alteration for each case, which cannot be clear from Figure 1. I suppose we should see 50% methylation level in the imprinted region for normal beta cell samples."*

Response: The Reviewer raises a good question that we should have addressed. There are two canonical imprinting control regions (ICRs) in our target region: the H19/IGF2 and the KvDMR1 ICRs. No statistically significant differentially methylated CpG dinucleotides were detected in these two ICRs. The average methylation values found for both ICRs are within normal ranges for ICR methylation, and are shown in the Table below:

chr	start	end	ICR	percent methylation	
				beta cells	insulinomas
chr11	2,018,812	2,024,740	H19/IGF2	49.5	47.3
chr11	2,719,948	2,722,259	KvDMR1	41.9	36.5

We have added this information to the second paragraph of the Discussion on page a 11. Thank you.

4. *"To propose a model in Figure 6, CTCF binding need to be examined to show the change of 3D looping. Most of the CTCF binding sites are shared among various cell types, but it would be ideal to examine the CTCF binding sites in EndoC-βH1 cells."*

Response: We also completely agree that CTCF ChIPseq is a critical next step, but we do not agree that EndoC-βH1 cells are a good choice for a "normal" control. They and the subsequent versions, βH2 and βH3, have been immortalized with Tert and T-antigen and do not reflect events in normal beta cells or human insulinomas. This is best illustrated by Stitzel et al (see Fig. 1 in *Cell Reports* <https://doi.org/10.1016/j.celrep.2018.12.083>) who

have shown that these cells contain large and multiple regions of chromosomal gain, loss and rearrangement. Thus, while we greatly admire the work involved in creating the immortalized EndoC-βH cell lines, and their utility as surrogate cell lines for drug screening, we believe they are no substitute for “normal human beta cells”. This is why we regularly employ FACS-sorted human beta cells from 10-100 human organ donors as the requisite normal control (see refs 3 and 4, as well as *Science Translational Medicine* PMID: 32051230).

Not surprisingly, CTCF ChIPseq has never been performed in human beta cells. As noted in response to question 2 above, we are limited with respect CTCF ChIPseq by cell number. We have been able to develop ChIPseq approaches for histone marks and ATACseq as described above, and are in the process of trying to develop CTCF ChIPseq using cut-and-run approaches. We have discussed this important limitation in the issue in the final paragraph of the revised Discussion on page 13.

5. “Given that there is heterogeneity in methylation status, you may need to compare the gene expression individually, not as a group.”

Response: We agree and had done just this in Supplementary Fig. 7 for the six samples for which both DNA methylation and gene expression data were available (see Supplementary Figure 1 and Supplementary Table 3 for details). We carried out a sample-by-sample comparison of promoter DNA methylation vs promoter gene expression (Supplementary Figure 7) which supports the findings in Figure 2A. Thank you.

“Minor points:”

- “In the heatmap of DNA methylation data in Figure 1B, they may get better clustering pattern by selecting a set of variable CpG sites.”

Response: Once again, we agree and had re-run the heatmap algorithm on those CpG dinucleotides with the highest variability in the course of the original manuscript preparation. The clustering pattern is similar to Fig. 1B, including all CpG dinucleotides. Below, we show the heatmap that displays the clustering pattern obtained by using the CpG dinucleotides in the top 10% (about 2,600 CpGs) for standard deviation across the whole sample set (5 beta cells and 19 insulinomas). We have not included this in the revision but would be happy to do so if the Reviewer and Editor feel strongly about this. Please advise. Thank you.

- “In Figure 2, they divided the genome into 100-kb bins and showed the percentage of chromatin status, which is too coarse to discuss promoter or enhancer position for gene-rich 11p15 regions.”

Response: We certainly agree that the bin size in Figure 2C (now Figure 3C) is coarse. Our goal was simply to provide a general, genome-wide, depiction of areas of high concentrations of promoter/enhancer/dyad sequences, as defined by the Roadmap Epigenomics consortium. For this purpose, reducing the bin size should not substantially change relative capture of promoter/enhancer/dyad sequences containing CpGs, as the smaller bin size would capture the same regions as those with high concentrations of promoter/enhancer/dyad sequences. This is illustrated in the adjacent Figure, which shows the effects of binning of the human genome into 10-kbp windows. This generated ~245,000 windows carrying different degrees of promoter/enhancer/dyad coverage. This number of datapoints cannot be graphed with the same approach used in Figure 2C. We therefore re-built the Figure as shown on the following page.

We have not included this figure in the revised manuscript since it does not alter the interpretation of the original Figure 2C (now 3C), but would be very happy to do so if the Editor or Reviewer feel that we should, perhaps as a Supplemental Figure. Again, please advise as to your preferences.

Figure Legend. The binning of the human genome into 10-kbp windows, as opposed to our previous 100-kbp binning. This generated ~245,000 windows carrying different degrees of promoter/enhancer/dyad coverage. This large number of datapoints is not graphically amenable to the approach used to build **Figure 2C**.

A. Genome-Wide Panel (left). 1. We calculated genome-wide per-window promoter/enhancer/dyad coverage. 2. We ranked all ~245,000 windows genome-wide for promoter/enhancer/dyad. 3. We selected the top 5th percentile of windows for promoter/enhancer/dyad coverage. 4. For each chromosome, we normalized the number of windows as calculated in point 3 to the total number of windows in that chromosome. 5. We graphed the percent distribution of the normalized windows across chromosomes. **Interpretation:** This approach identified chr11 as the 4th most enriched chromosome for windows in the top 5th percentile for enhancer coverage, similar to the original **Figure 2C**.

B. 11p15.5-p15.4 Target Subregion Panel (right). Here, for each set of sequences (promoters, enhancers and dyads), we graphed per-window coverage genome-wide rank across the 11p15.5-p15.4 target subregion. **Interpretation:** A strong enrichment signal can be detected for enhancers with elevated genome-wide rank across the 11p15.5-p15.4 target subregion (mean rank = 82, scored windows = 131). In contrast, this enrichment is not observed for promoters (mean rank = 52, scored windows = 30) or dyads (mean rank = 53, scored windows = 63). This panel confirms the original interpretation of the insert charts at the top of **Figure 2C**.

Once again, we want to express our deep appreciation to all four Reviewers for their invaluable efforts to improve the manuscript. We believe, and hope they will agree, that this is an important contribution to the insulinoma, Beckwith-Weidemann, Focal Hyperinsulinism as well as pancreatic neuroendocrine tumor literature.

REVIEWERS' COMMENTS:

Reviewer #2 (Remarks to the Author):

All questions have been satisfactorily answered.

Reviewer #3 (Remarks to the Author):

Thanks for clarifying that the results obtained from Fig 1B were not then used to generate further analyses of the manuscript.

In this case I would recommend removing Fig 1 B.

I do not have further comments.

Reviewer #4 (Remarks to the Author):

In the revised manuscript, although the authors mostly agreed with the comments, sufficient changes were not made accordingly.

1a. I am still puzzled why they kept the text 525-531 regarding NKX3-1 in Discussion. As I previously commented, NKX2-2 is well known as a regulator of the beta cell development, while NKX3-1 expression level is quite low in this study. They also stated that they do not believe that NKX3-1 binding loci play a role in beta cell or insulinoma biology, this discussion is misleading. They should rather discuss the role of NKX2-2, instead. They should not list NKX3-1 in Figures 6B and 7C.

1b. Gene expression would be useful in predicting the responsible gene among family genes. In Supplementary Table 16 they should list NKX2-2 and NFATC2 too, as both genes are highly expressed in insulinoma.

2. As cited in a newly added Reference #36, overexpression of either Nfatc2 or Nfatc1 in mouse islets enhanced insulin secretion, whereas only Nfatc2 was able to promote β -cell proliferation. Together with the genetic data, such as eQTL, in Ref #36, Nfatc2 is involved in beta cell biology.

Unless they do have any evidence NFATC1 is more important in insulinoma biology than NFATC2, they should not list NFATC1 in Figures 6B, 7C and 8. To be accurate, they can show the transcription factor binding motif in Figures 6B and 7C.

I understand that availability of appropriate cell lines and clinical samples was a problem for them to perform epigenomic analysis, such as ChIP-seq. Alternatively, immunohistochemical analysis of NFATC proteins could be performed on insulinoma specimens to determine the responsible factor.

3. No further comment.

4. No evidence to propose the model in Figure 8. They could discuss the direction of CTCF binding motifs.

5. No further comment.

Minor points

- Methylation cluster

I do not agree with their opinion that the methylation clustering pattern with variable probes is similar to that in the original Fig. 1B with all genes. In the original figure, ins_18 sample was clustered together with 5 beta cell samples, while in the reanalyzed data it was together with insulinoma Group 2. I would suggest Fig. 1B should be replaced with the new figure.

It is also interesting that insulinoma samples were mostly divided into 2 groups, Group 1 and Group 2, by DNA methylation. They should discuss the differentially methylated probes as classifiers. I wonder if they could find any association of clinical parameters with subgroups.

- Figure 3 (previous Figure 2)

Genome wide data is not required for this paper. In 10kb window figure, we can clearly see the change of DNA methylation level, compared to the Figure 3C(new). Newly prepared Figure panel B (right) could be useful, so should replace Figure 3C.

Response to Reviewers

Thank you for your comments on our revised manuscript. We appreciate your and the Reviewers' careful and thoughtful review, and the rapid turnaround. Below, in blue text, please see our responses to the Reviewers' questions and comments. Please particularly note the response to Reviewer 3, in which we describe the combinations of Figs 1 and 2 into a new Fig 1, which necessitates revising the remaining Figure numbers and their citations in the text.

REVIEWERS' COMMENTS:

Reviewer #2 (Remarks to the Author):

"All questions have been satisfactorily answered."

Response: No response required. Thank you.

Reviewer #3 (Remarks to the Author):

"Thanks for clarifying that the results obtained from Fig 1B were not then used to generate further analyses of the manuscript. In this case I would recommend removing Fig 1B. I do not have further comments."

Response: Reviewer 4 also commented on Figure 1B in item 6 below. Briefly, Reviewer 3 wants to remove Fig 1B, and Reviewer 4 wants to keep the revised Fig 1B. Reviewers 1 and 2 had no issues with Fig 1B. We have chosen what we believe is a reasonable compromise for Reviewers 1- 4: we have removed Fig 1B from the main figures (for Reviewer 3), made it a new Supplementary Fig 3, and added the revised version of Fig 1B as a new Supplementary Figure 6 (for Reviewer 4). The overall effect of these changes is that it removes any suggestion of statistically significant sub-clusters of insulinomas (for Reviewers 3 and 4, and we also agree), yet also makes the point that no matter how one chooses to assess the insulinoma subtypes, their methylation profiles are remarkably consistent, and very different from normal beta cells.

These changes required combining figures in the last version as follows: Figs 1 and 2 have been combined to create a new Figure 1A,B,C. Subsequent Figures have been re-numbered accordingly, and their citations in the text have been revised accordingly. Thank you.

Reviewer #4 (Remarks to the Author):

"In the revised manuscript, although the authors mostly agreed with the comments, sufficient changes were not made accordingly."

1a. "I am still puzzled why they kept the text 525-531 regarding NKX3-1 in Discussion. As I previously commented, NKX2-2 is well known as a regulator of the beta cell development, while NKX3-1 expression level is quite low in this study. They also stated that they do not believe that NKX3-1 binding loci play a role in beta cell or insulinoma biology, this discussion is misleading. They should rather discuss the role of NKX2-2, instead. They should not list NKX3-1 in Figures 6B and 7C."

Response: We certainly agree with the Reviewer that the NKX3-1 finding is unexpected. We also of course agree that the transcription factor NKX2-2 is recognized as a critical regulator of pancreatic endocrine cell differentiation, whereas NKX3-1 is not. We also agree that NKX3-1 is expressed at lower levels - although still biologically important (34th percentile of 18,664 expressed beta cell genes in our prior *Nature Communications* report (ref 10)) - than NKX2-2. On the other hand, NKX3-1 emerges from a pure data-driven approach, seeking transcription factors that fit two criteria: their binding sites are hypermethylated, and they are also overexpressed. Simply stated, NKX3-1 fits these criteria and NKX2-2 does not. We believe the reason that NKX3-1 appears in the analysis, rather than NKX2-2, is that NKX3-1 is differentially expressed in beta cells vs. insulinomas, whereas NKX2-2 is equally expressed. We hope the Reviewer will agree that ignoring NKX3-1 and/or substituting NKX2-2 because of preconceptions would be unacceptable. Finally, we note that binding sites for NKX2-2 and NKX3-1 partly overlap; it is thus possible that NKX3-1 differential methylation emerges because of this. With the Reviewer's comment in mind we have added additional language to the Discussion on page 12, paragraph 3 discussing these issues. Thank you.

1b. Gene expression would be useful in predicting the responsible gene among family genes. In Supplementary Table 16 they should list NKX2-2 and NFATC2 too, as both genes are highly expressed in insulinoma."

Response: Again, the same issue applies. We were looking in this analysis for transcription factors that were differentially expressed and whose binding sites were differentially methylated. NFATC2 is not differentially expressed in beta cells vs. insulinomas and thus does not fit these criteria. This does not mean that NFATC1 is more or less important than NFATC2 in beta cell and insulinoma biology: we are just reporting the findings. See the response to the following question as well. Thank you.

2. "As cited in a newly added Reference #36 (note, this is ref 31), overexpression of either Nfatc2 or Nfatc1 in mouse islets enhanced insulin secretion, whereas only Nfatc2 was able to promote β -cell proliferation. Together with the genetic data, such as eQTL, in Ref #31, Nfatc2 is involved in beta cell biology. Unless they do have any evidence NFATC1 is more important in insulinoma biology than NFATC2, they should not list NFATC1 in Figures 6B, 7C and 8. To be accurate, they can show the transcription factor binding motif in Figures 6B and 7C."

Response: We actually do not have a preference for NFATC1 vs. NFATC2. Again, we are simply reporting the unbiased results of the analysis. We understand that the Reviewer favors Nfatc2 over Nfatc1, but offer the following counterpoints. First, the data in ref 31 were obtained in juvenile mouse islets, which mimic adult human beta cells in some cases but diverge in many others, as the Reviewer must certainly agree. Second, the Nfatc1/2 studies in ref 31 were overexpression studies which may - or may not - reflect physiological events in human beta cells. Third, the constructs employed were mutant, constitutively active Nfatc1/2 isoforms which may - or may not - faithfully reflect events with wild-type Nfatc1/2 or human NFATC1/2. And fourth, the readout for proliferation was ³H-thymidine incorporation in whole islets which captures proliferation in all mouse islet cell types (endothelial, alpha, delta, fibroblast, ductal cells etc.) in addition to beta cells. Thus, our sense is that NFATC's are important, but there is no basis to prefer or recognize one over the other. For this reason, we have left the figures as they were, since NFATC1 emerged in an unbiased manner from the analysis, but have added a discussion of the NFATC1/2 issue to the Discussion on page 12, paragraph 4. Thank you.

I understand that availability of appropriate cell lines and clinical samples was a problem for them to perform epigenomic analysis, such as ChIP-seq. Alternatively, immunohistochemical analysis of NFATC proteins could be performed on insulinoma specimens to determine the responsible factor."

Response: Thank you for understanding the ChIPseq and cell line problems. We also definitely considered examining all four NFaT family members by immunohistochemistry in insulinomas as we have done in normal human beta cells (*Nature Medicine* PMID 25751815, 2015, Suppl. Fig. 5). We learned from that experience, as one might expect, that while all four NFaTs could be detected and validated in human beta cells, using uniform and gentle fixation techniques (0.1% paraformaldehyde, for 10 min, on dispersed cells), immunohistochemistry is not a reliable quantitative method for overfixed (4% formalin for variable times: days, weeks), randomly processed (in multiple surgical pathology departments) insulinoma specimens. In addition, it is possible, and perhaps likely, that different insulinomas may express different NFaT family proteins. With these thoughts in mind, we have not pursued NFaT immunohistochemistry in insulinomas. We also note, as we had earlier, that mitogenic events in rodent beta cells translate unpredictably to human beta cells. We perhaps could overexpress the four NFaTs in human islets and assess insulin secretion and proliferation. We hope the Reviewer will agree that this is best pursued in a subsequent study. Thank you.

3. "No further comment."

Response: No response required. Thank you.

4. "No evidence to propose the model in Figure 8. They could discuss the direction of CTCF binding motifs."

Response: As we make clear in the manuscript, and in Fig. 8 (now Figure 7), we provide a hypothetical model for insulinomas, on which to base future studies. The model is derived from unbiased data in the manuscript combined with prior 3-D chromatin structures of the insulin locus in beta cells. We believe, as did the prior three Reviewers, that it is a reasonable working model, and may very well be modified with time.

With regard to the question of CTCF binding motif directionality, we agree that this is an interesting question, and is important for the establishment/inhibition of DNA loop formation. We do believe, based on the studies of Felsenfeld et al and Stitzel et al (refs 12, 21,25,26) that there are interactions between the CTCF binding sites within the INS-IGF2 locus in beta cells highlighted in Figure 8B (new Fig 7B), upper panel. The interactions in insulinomas depicted in Fig. 8B, lower panel, between the INS-IGF2 locus and the SYT8/TNNI2 locus are hypothetical, as clearly indicated in the text and figures. Indeed, there are many interesting additional possibilities here: the 11p15.5-p15.4 target subregion carries some 331 differentially methylated CTCF binding sites, 13 of which reside within the INS-IGF2 locus, with different orientations, that likely permit

multiple different DNA looping interactions. These hypotheses will require additional investigation via future 3-D chromatin structural studies as we have indicated in the Discussion on page 13, final paragraph. Thank you.

5. "No further comment."

Response: No response required. Thank you.

Minor points

6a. – "Methylation cluster: I do not agree with their opinion that the methylation clustering pattern with variable probes is similar to that in the original Fig. 1B with all genes. In the original figure, ins_18 sample was clustered together with 5 beta cell samples, while in the reanalyzed data it was together with insulinoma Group 2. I would suggest Fig. 1B should be replaced with the new figure."

Response: Reviewer 3 also commented on Fig. 1B as noted above. Please see our response to Reviewer 3 above. Thank you.

6b. "It is also interesting that insulinoma samples were mostly divided into 2 groups, Group 1 and Group 2, by DNA methylation. They should discuss the differentially methylated probes as classifiers. I wonder if they could find any association of clinical parameters with subgroups."

Response: Please note that we have tried to de-emphasize the cluster analysis as described in response to Reviewer 3, and have moved subgroup illustrations to Supplementary Figures 3-7.

We are unclear what the Reviewer means by "differentially methylated probes" and classifiers: there were no probes. We used ~10,000 primer pairs to amplify *all* of the potentially methylated CpGs in the 11p15.5-p15.4 target subregion in beta cells and insulinomas and performed unbiased next-gen bisulfite sequencing of the entire region. All samples were treated in the same manner and sequenced in two batches that contained a mixture of beta cells and insulinomas. We do not think there is any concern for bias here.

With regard to clinical parameters, we did look for differences in clinical parameters among the insulinoma groups. There were no differences in mean age, sex, lowest fasting blood glucose, or insulin level, as indicated in Supplementary Table 2. Nonetheless, the Reviewer is correct that more subtle differences may have been present, for example differences in the glucose threshold for insulin suppression, but it was not possible to assess these parameters in this cohort of rare insulinomas received on short notice from many institutions. On the other hand, since we have gone to great pains to address the Reviewers' desire that we not imply or suggest that there are subgroups of insulinomas defined by methylome profiles, we have not altered the text to discuss clinical differences in the "subgroups)". We hope the Reviewer will agree that this is reasonable.

7. – "Figure 3 (previous Figure 2) Genome wide data is not required for this paper. In 10kb window figure, we can clearly see the change of DNA methylation level, compared to the Figure 3C(new). Newly prepared Figure panel B (right) could be useful, so should replace Figure 3C."

Response: We agree that this is reasonable, and Figure 3 (now Figure 2C) has been revised accordingly, along with its Figure Legend. Thank you.

Once again, we want to thank the Reviewers for their interest, thoughts and suggestions.